# Inversion of pheromone preference optimizes foraging in *C. elegans*

**Martina Dal Bello[1]\*, Alfonso Pérez-Escudero[1,2], Frank C Schroeder[3], Jeff Gore[1]\***

[1]Physics of Living Systems Group, Department of Physics, Massachusetts Institute of Technology, Cambridge, United States; [2]Centre de Recherches sur la Cognition Animale (CRCA), Centre de Biologie Intégrative (CBI), Université de Toulouse; CNRS; UPS, Toulouse, France; [3]Boyce Thompson Institute and Department of Chemistry and Chemical Biology, Cornell University, New York, United States

**Abstract** Foraging animals have to locate food sources that are usually patchily distributed and subject to competition. Deciding when to leave a food patch is challenging and requires the animal to integrate information about food availability with cues signaling the presence of other individuals (e.g., pheromones). To study how social information transmitted via pheromones can aid foraging decisions, we investigated the behavioral responses of the model animal *Caenorhabditis elegans* to food depletion and pheromone accumulation in food patches. We experimentally show that animals consuming a food patch leave it at different times and that the leaving time affects the animal preference for its pheromones. In particular, worms leaving early are attracted to their pheromones, while worms leaving later are repelled by them. We further demonstrate that the inversion from attraction to repulsion depends on associative learning and, by implementing a simple model, we highlight that it is an adaptive solution to optimize food intake during foraging.

## Introduction

Foraging for food is among the most critical activities for an animal's survival (*Calhoun et al., 2014*). It is also among the most challenging, because food is usually patchily distributed in space and time, and other individuals are attempting to find and consume the same resources (*Abu Baker and Brown, 2014*; *Driessen and Bernstein, 1999*; *Stephens and Krebs, 1987*).

An important factor, which has been the focus of considerable effort in models of foraging behavior, is for how long to exploit a food patch. At any given time, an individual feeding in a food patch has to choose between leaving to search for a better patch or staying. Leaving incurs the cost of exploring an unknown territory, while staying results in the cost of feeding in a depleting food patch. Most models addressing this 'dilemma' involve patch assessment by individuals and postulate that the leaving time depends on local estimates of foraging success (*Charnov, 1976*; *Oaten, 1977*; *Stephens and Krebs, 1987*). As such, foragers are predicted to depart from a food patch when the instantaneous intake rate drops below the average intake rate expected from the environment, a phenomenon that has been observed in several animals, from insects (*Wajnberg et al., 2008*) to birds (*Cowie, 1977*; *Krebs et al., 1974*) and large mammals (*Searle et al., 2005*). The presence of other animals, however, affects individual foraging success so that different leaving times can be expected (*Aubert-Kato et al., 2015*; *Couzin et al., 2005*; *Giraldeau and Caraco, 2000*; *Karpas et al., 2017*).

Once an animal leaves a food patch, it will have to explore the environment to locate new sources of food. Since natural habitats are usually saturated with many different non-specific chemical cues, animals use pheromones and other odors to orientate their searches (*Wyatt, 2014*). This, however, implies determining whether pheromones point toward a resource supporting growth and reproduction or an already exploited one. To acquire this knowledge, animals have to learn from experience.

**\*For correspondence:**
dalbello@mit.edu (MDB);
gore@mit.edu (JG)

**Competing interests:** The authors declare that no competing interests exist.

In the context of social foraging, it has been shown that individuals might need to rely only on the most recent experience (*Krebs and Inman, 1992*). As such, the valence (positive or negative signal) of pheromones acquired during the most recent feeding activity is crucial for the success of the foraging process. While this has been shown in bumblebees feeding on transient resources (*Ayasse and Jarau, 2014*), we still do not know whether it is important for other animals feeding in groups. Moreover, it is not clear if the ability to use associative learning—the capacity to learn and remember the features of the environment that are associated with positive or aversive stimuli (*Ardiel and Rankin, 2010*)—to change the valence of pheromones could improve foraging success.

The nematode *Caenorhabditis elegans* is a powerful model system to investigate how information about food availability and pheromones can shape foraging in patchy habitats. *C. elegans* feeds in large groups on ephemeral bacterial patches growing on decomposing plant material, a habitat that can be mimicked in a petri dish (*Frézal and Félix, 2015*; *Schulenburg and Félix, 2017*). Importantly, *C. elegans* can evaluate population density inside food patches using a suite of pheromones, belonging to the family of ascarosides, which are continuously excreted by worms (*Greene et al., 2016*; *Ludewig and Schroeder, 2013*). Finally, it has been shown that pheromones and food availability control the leaving times of foraging worms. In particular, the rate at which individuals abandon the patch increases when food becomes scarce and pheromones are at high concentrations (*Figure 1A*; *Harvey, 2009*; *Milward et al., 2011*).

In the present study, we experimentally investigated the behavioral responses of *C. elegans* to food depletion and pheromone accumulation in food patches. We confirmed that individual worms consuming a food patch leave at different times, and we found that worms leaving early have a preference for worm-secreted pheromones while those leaving late avoid the pheromones. A simple foraging model suggests that these two behaviors may optimize foraging success in the presence of competitors. Finally, using a series of behavioral assays altering worm exposure to food and pheromones, we demonstrate that associative learning underpins the change in pheromone preference.

## Results

### The leaving time from a food patch affects the preference for pheromones of a *C. elegans* natural isolate

To investigate *C. elegans* behavioral responses to food depletion, we developed an assay to simultaneously assess patch-leaving behavior and pheromone preference over time (see Materials and methods section and *Figure 1B*). We used young adult hermaphrodites of the natural isolate MY1 (Lingen, Germany) to assess behavioral patterns that could be as close as possible to those exhibited by *C. elegans* in its natural habitat. In our assay, a small patch of bacteria is gradually depleted by feeding animals (about 5 hr). At equal distances from the food patch there are two spots, one of which contains a pheromone blend. Shortly after worms leave the food patch, they make a decision by choosing between the two spots (*Figure 1—figure supplement 1*). This assay mimics a foraging activity in which animals decide first when to leave a food patch and later whether to follow a pheromone cue, which is indicative of the presence of others. In these experiments, the pheromone blend is obtained by collecting and filtering the supernatant of well-fed worms maintained in a liquid culture (*Choe et al., 2012*; *Harvey, 2009*; *White et al., 2007*). In agreement with previous results (*Milward et al., 2011*), we found that feeding animals abandon the food patch at very different times, with some worms leaving at the beginning while others staying until the food patch is depleted (*Figure 1C*). In addition, the leaving time affects worms' preference for their pheromones, with individuals leaving the food patch early (first three hours) going to the pheromone blend (positive chemotaxis index, *Figure 1D*) while worms leaving the food patch later avoiding it (negative chemotaxis index, *Figure 1D*). *C. elegans* responses to food depletion, therefore, include an inversion in pheromone preference dependent on the leaving time from the food patch.

### A simple model shows that the inversion of the preference for pheromones can be a strategy to optimize foraging

Inside rotting fruits and stems where *C. elegans* forage, bacterial food is patchily distributed (*Frézal and Félix, 2015*; *Schulenburg and Félix, 2017*). We might then expect that the timing of dispersal from existing food patches and the strategies that optimize food intake are crucial for

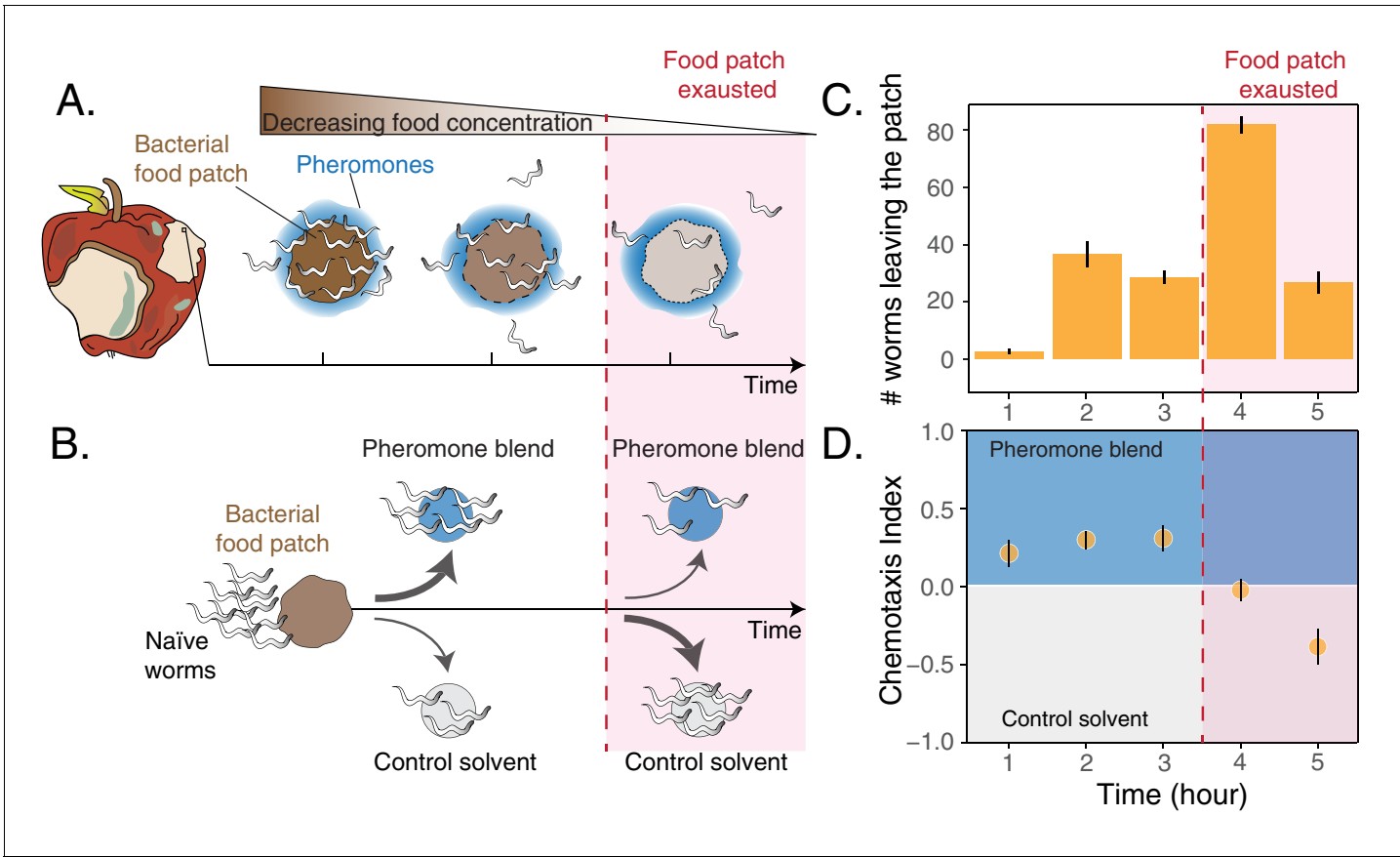

**Figure 1.** Worms leaving at different times from a food patch exhibit opposite preferences for pheromones. (**A**) During the feeding process, worms remaining in the food patch experience different environmental conditions. At the beginning, food is still abundant and pheromones have already accumulated. By the end, food is scarce and pheromone concentration is even higher. (**B**) In the behavioral assay, as animals feed and leave from a food patch, they are presented with the choice between a spot containing the pheromone blend and a spot containing a control solvent. In the two spots, sodium azide is added in order to anesthetize the animals and prevent them from leaving the chosen spot. (**C**) Individual worms leave the food patch at different times. The average number of worms that abandoned the food patch at each hour is shown (mean worm count ± SEM across replicates, n. experiments = 2). (**D**) Animals leaving the food patch earlier prefer the pheromone blend while those leaving later, when the food is almost depleted, avoid the pheromone blend. In the plot, chemotaxis index is calculated on the number of naïve MY1 young adult hermaphrodites that, at each hour, reach the two spots (mean CI ± SEM across replicates, n. experiments = 2). The red region in each plot approximately indicates when food in the patch is exhausted.

The online version of this article includes the following source data and figure supplement(s) for figure 1:

**Source data 1.** Choice after food assay data.
**Figure supplement 1.** Layout of plates for chemotaxis assays.

worm survival. A natural question then arises: can the behaviors we observed in our experiments provide any benefit to *C. elegans* foraging? We addressed this question with a simple theoretical model exploring the optimality of the inversion in pheromone preference in the context of foraging in a heterogeneous environment. This model uses the tools of Game Theory to find the strategy that maximizes the food eaten by a worm, taking into account that other worms will also follow the same strategy. This strategy is called Evolutionary Stable Strategy, and should have been selected by evolution (*Maynard Smith, 1982*).

Our model considers one unoccupied and two occupied food patches (*Figure 2A*). Initially, $n_1$ worms are in patch 1, and $n_2$ worms are in patch 2 (*Figure 2A*). We assume by convention that $n_1 \geq n_2$, so patch 1 is initially overcrowded and patch 2 is undercrowded. Worms have three possible choices: (1) *remain* in their current food patch, (2) *switch* to another occupied food patch, and (3) *disperse* away from the occupied patches, in search for an unoccupied one. *Switching* means that a worm will leave its initial food patch and follow pheromone cues in order to find another

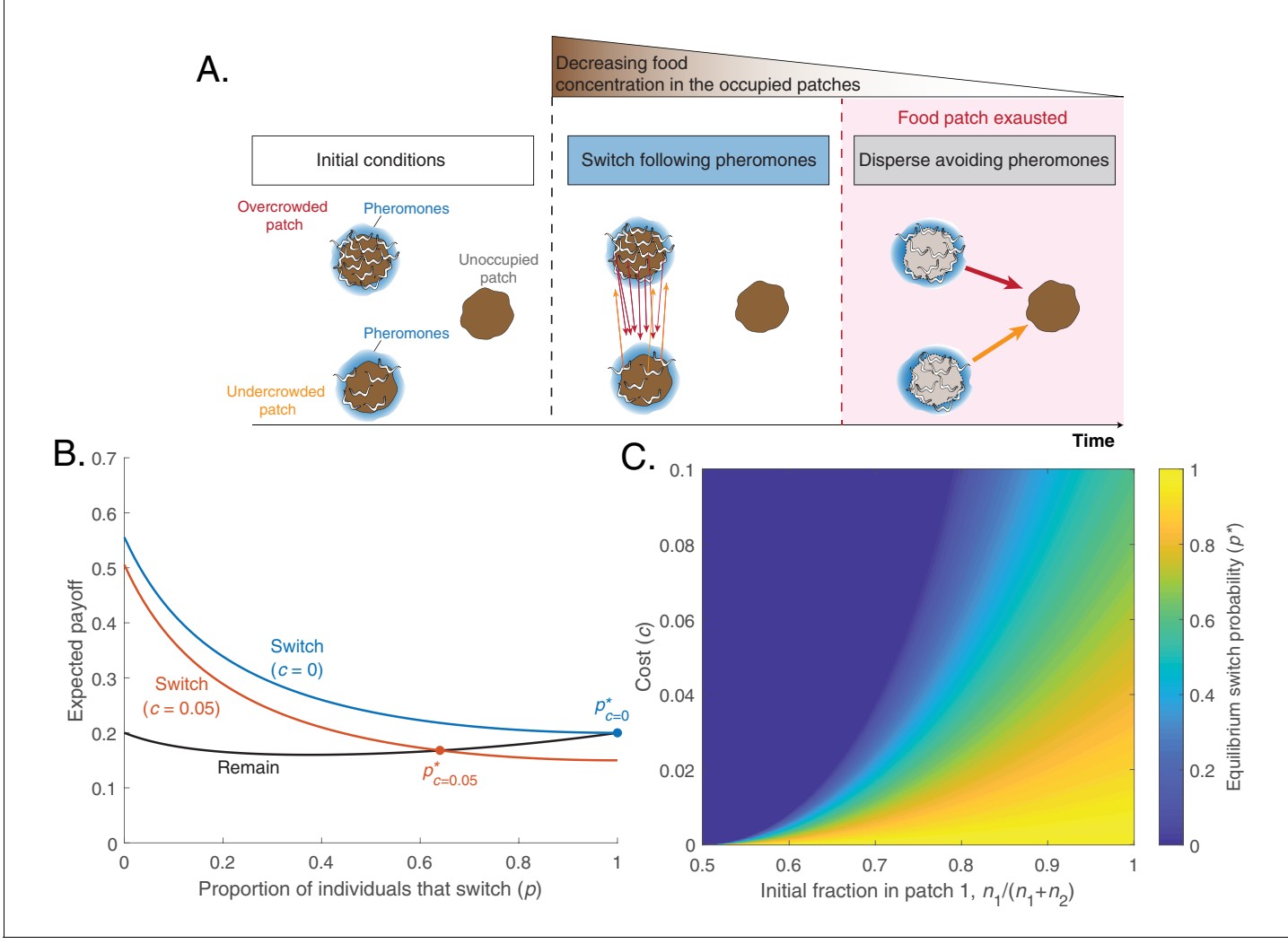

**Figure 2.** A simple model predicts that worms can optimize foraging by a change in pheromone preference over time. (A) Schematic of model predictions (only three identical food patches are depicted). Initially, two patches are unequally populated by *C. elegans* individuals (overcrowded and undercrowded food patch) while a third one is unoccupied. The release of pheromones by worms makes the occupied patches easier to find compared to the unoccupied one. During the first phase, worms equalize occupancy in the occupied patches. Then, all worms stay in their patches until food becomes scarce. In this last phase, worms benefit from dispersing to the unoccupied patch avoiding pheromone cues. This would be favored by a change in pheromone preference. (B) Expected payoff for each strategy (Remain, $\langle H \rangle_{\text{Remain}}$, in black; Switch $\langle H \rangle_{\text{Switch}}$, in color as a function of the fraction of individuals that switch ($p$). In the absence of cost, switching is always advantageous (blue line), so the equilibrium switching probability is $p^*_{c=0} = 1$ (blue dot). A switching cost shifts the equilibrium probability to an intermediate value (orange dot). Lines computed with **Equations 3a and 3b** with parameters $n_1 = 9$, $n_2 = 1$, $A_E = 1$. (C) Probability to switch in equilibrium ($p^*$), as a function of switching cost c and the initial fraction of worms in patch 1($\frac{n_1}{n_1+n_2}$)).

The online version of this article includes the following figure supplement(s) for figure 2:

**Figure supplement 1.** Benefit that individuals obtain from switching at time $t = 0$, with respect to individuals that remain , versus the probability of switching.

**Figure supplement 2.** Benefit that individuals obtain from switching at time $t = 0$, with respect to individuals that remain, versus the probability of switching when patches have equal or different sizes.

occupied food patch. *Dispersing* means that the worm will leave its current food patch and avoid pheromones to maximize the probability of finding an unoccupied food patch. We assume that unoccupied food patches are hard to find, because they are not marked by pheromones and may be on average further away. Therefore, dispersal will not be advantageous until the occupied food patches are nearly depleted. A proof of this result, which closely resembles the marginal value theorem (*Charnov, 1976*), can be found in the supplement; here we will simply assume that worms will

not *disperse* until the occupied food patches are depleted. Therefore, initially the individuals will choose between *remaining* or *switching*.

We assess now the consequences of switching. We assume that the choice between switching or remaining cannot depend on the worm's initial patch, because worms cannot know whether they start in the overcrowded or the undercrowded patch. Indeed, even if an individual can assess the density of its conspecifics in its current food patch, it does not have reliable information about the density of the other patch. As such, all worms have a probability $p$ of switching, regardless of their initial patch. Also, worms that decide to switch may return to their original patch before finding the other one, so worms that switch have a probability ½ of ending up in either food patch. Therefore, the number of worms in the $i$-th food patch after the switch will be

$$m_i = n_i(1-p) + \frac{(n_1+n_2)p}{2},$$ (1)

where the first term represents the worms that started in the $i$-th patch and remained there, and the second term represents the worms that switched and ended up in the $i$-th patch (regardless of their initial patch). After the switch, worms exploit their chosen food patch until it's exhausted. The food in each patch is shared evenly among the worms exploiting it, so if each food patch contains $A_E$ units of food, each worm in the $i$-th patch will eat

$$G_i = \frac{A_E}{m_i} = \frac{A_E}{n_i(1-p) + \frac{(n_1+n_2)p}{2}}$$ (2)

units of food. We can now ask: Is switching on average more advantageous than remaining? Does it depend on the cost of switching or on the initial distribution of worms in the two food patches?

To test if switching is more advantageous than remaining, we calculate the payoff for each strategy. The payoff depends on the food intake achieved in each patch (*Equation 2*) and the probability for each worm to end up in either of the two occupied food patches after switching or remaining. We assume that worms that switch have a probability ½ of ending in either food patch, while worms that remain have a probability $\frac{n_i}{n_1+n_2}$ of being in the $i$-th food patch. The expected payoff for *remaining* then is

$$\langle H \rangle_{\text{Remain}} = \sum_{i=1}^{2} \frac{n_i}{n_1+n_2} \frac{A}{n_i(1-p) + \frac{(n_1+n_2)p}{2}}.$$ (3a)

The payoff for *switching* is instead

$$\langle H \rangle_{\text{Switch}} = \sum_{i=1}^{2} \frac{1}{2} \frac{A}{n_i(1-p) + \frac{(n_1+n_2)p}{2}} - c,$$ (3b)

where $c$ is the cost of switching.

If both the switching probability and the cost are low ($p \approx 0, c \approx 0$), the expected payoff is higher for switching than for remaining. This happens because every individual has a higher probability of being in the overcrowded patch than in the undercrowded one and therefore has an incentive to switch. The difference between the two payoffs diminishes as the probability of switching increases, but if switching is costless all worms should switch in order to perfectly equalize their distribution across the two food patches (*Figure 2B*, blue). In contrast, when switching is costly, the equilibrium is reached when only a fraction of the population switches (*Figure 2B* orange). This equilibrium fraction ($p^*$) depends both on the cost of switching and on the initial imbalance in patch occupancy (*Figure 2C*).

Our model recapitulates the two key experimental observations highlighted in *Figure 1*: First, a fraction of worms will switch at the beginning, leaving the food patch before it is depleted and following pheromones to reach another occupied food patch. Second, once the food patches are depleted all worms will disperse, avoiding depleted food patches by reversing their preference for pheromones (which now mark depleted food patches). Here, we have illustrated these results with a simplified model, with two identical food patches and in which worms can only switch or disperse at particular times. A more general model in which individuals can move at any time between any

number of food patches of equal or different sizes gives the same Evolutionary Stable Strategy (see Appendix 1).

Our theoretical results show what features of *C. elegans* environment may lead to the evolution of the observed behaviors, regardless of how the behaviors are implemented. In particular, the inversion in pheromone preference may be triggered by several different factors, and our model cannot distinguish between them. In the following, we will examine experimental evidence related to these mechanisms.

## The change in pheromones preference is likely due to associative learning

As anticipated, our model does not encode any specific mechanism underpinning the inversion of pheromone preference. The most parsimonious explanation is that animals leaving the patch earlier might differ from worms leaving later simply due to their feeding status. Indeed, early-leaving worms abandon the food patch when food is still abundant and therefore, they are more likely to be well-fed. By contrast, worms leaving later—when the food is scarce—are more likely to be famished. However, worms leaving earlier are also exposed to pheromones in the presence of abundant food, while worms leaving later experience high levels of pheromones in association with scarce food. These conditions are analogous to those that have been shown to support associative learning in *C. elegans* (*Ardiel and Rankin, 2010*). Similarly to the well-known case of associative learning with salt (*Hukema et al., 2008*; *Saeki et al., 2001*), in our experiment worms could be initially attracted to pheromones because of the positive association with the presence of food. Attraction could later turn into repulsion if worms start associating pheromones with food scarcity.

To distinguish between the change in pheromone preference being caused by feeding status alone or by associative learning, we performed experiments in which young adult hermaphrodites were conditioned for five hours in the four scenarios corresponding to the combinations of $\pm$ *food* and $\pm$ *pheromone blend* (see Conditioning experiments in the Materials and methods section). After conditioning, animals were assayed for chemotaxis to the pheromone blend (*Figure 3A*). We found that worms go to the pheromone blend when they are conditioned with *+food + pheromone blend* whereas they avoid it when they are conditioned with *– food +pheromone blend* (*Figure 3B*, blue and yellow bars, chemotaxis index of $CI_{++} = 0.38 \pm 0.07$ vs $CI_{-+} = -0.15 \pm 0.04$). Interestingly, worms conditioned without the pheromone blend do not exhibit a particular preference for their pheromones (*Figure 3B*, red and turquoise bars, chemotaxis index for the *+ food – pheromone blend* and the *– food – pheromone blend* scenarios are $0 \pm 0.07$ and $-0.02 \pm 0.06$, respectively). Worms therefore exhibit attraction when pheromones are paired with abundant food and aversion when pheromones are associated with absence of food. Otherwise, *C. elegans* does not show a specific preference for pheromones. These findings are consistent with the hypothesis that the *C. elegans* preference for the pheromone blend changes due to associative learning.

Conditioning in the two scenarios without pheromone blend added had to be performed at low worm density due to uncontrolled pheromone accumulation. Indeed, when conditioned at high worm density, animals in the *+food – pheromone blend* scenario are still exposed to the pheromones that they keep excreting during the 5 hr conditioning period (*Sakai et al., 2013*) and therefore exhibit attraction to the pheromone blend, albeit variable ($CI_{+-} = 0.25 \pm 0.07$, *Figure 3B* inset, red bar). Worms conditioned in *– food – pheromone blend* display no significant attraction to the pheromone ($CI_{--}=0.10 \pm 0.15$, *Figure 3B* inset, turquoise bar). The variation here is even bigger, likely due to the fact that the pheromone cocktail produced by starved worms can be different from the pheromone blend we used, which was obtained from well-fed worms (*Kaplan et al., 2011*).

In addition to a cocktail of ascaroside pheromones, the pheromone blend contains other products of worm metabolism, compounds deriving from the decomposition of dead worms and bacteria, and perhaps other unknown substances. Worms could in principle learn and change preference for any of these compounds and thus forage efficiently as indicated by our model. To probe whether specific pheromones are involved in the foraging optimization, we asked two questions: 1. Can *C. elegans* attraction to ascarosides be turned into repulsion? 2. Can *C. elegans* learn with ascarosides? To address these questions, we conducted the conditioning experiments (see Materials and methods) with two synthetic ascarosides, ascr#5 and icas#9, instead of the pheromone blend. We found that *C. elegans* can change its preference for both ascarosides (*Figure 3—figure supplement 1*), suggesting that ascaroside pheromones are likely contributing to the response to the pheromone

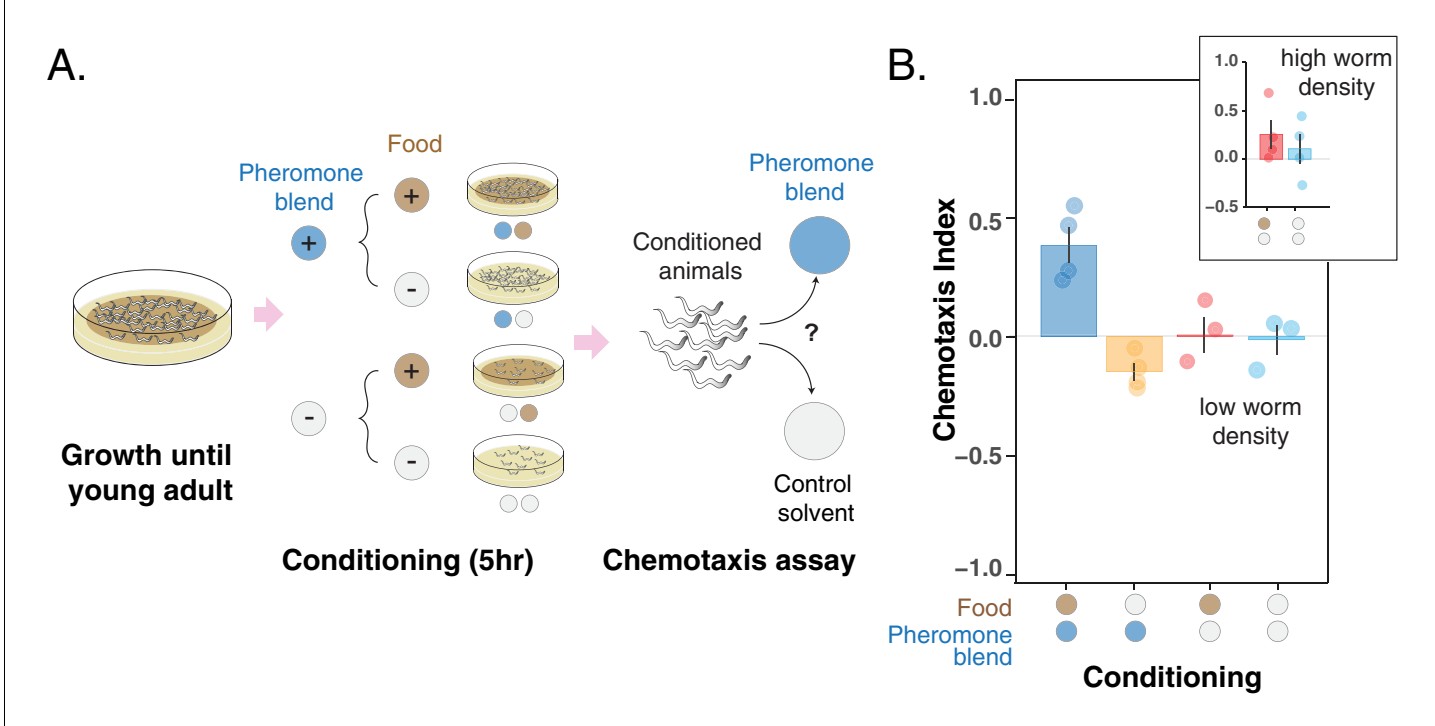

**Figure 3.** Changes in pheromone preference depend upon associative learning. (**A**) MY1 individuals grow at high density and with plenty of food until young adult. Animals are then transferred to conditioning plates, where they spend 5 hr. Conditioning scenarios are four: + *food + pheromone blend;* – *food + pheromone blend; + food – pheromone blend; – food – pheromone blend.* To prevent uncontrolled pheromone accumulation, the conditioning scenarios without added pheromone blend had to be repeated at low worm density. Worms are then assayed for chemotaxis to the pheromone blend. (**B**) MY1 individuals are not generally attracted the pheromone blend unless it is paired with abundant food. Chemotaxis index is shown for the four different conditioning scenarios: + *food + pheromone blend* (blue bar); – *food + pheromone blend* (yellow bar); + *food – pheromone blend* (red bar); – *food – pheromone blend* (turquoise bar). As a comparison, chemotaxis index is shown for the + *food – pheromone blend* scenario and the – *food – pheromone blend* scenarios with conditioning done at normal animal density (Panel B, inset). Points indicate the outcome of each independent replicated experiments (n = 4 and n = 3 for experiments with worms at low population density) while bars indicate the CI ± SEM across independent experiments.

The online version of this article includes the following source data and figure supplement(s) for figure 3:

**Source data 1.** Conditioning with pheromones data.

**Figure supplement 1.** Attraction toward two ascarosides, ascr#5 and icas#9, can be turned into repulsion.

**Figure supplement 1—source data 1.** Conditioning with ascarosides data.

blend found previously. However, these experiments were performed at high worm density, meaning that accumulation of secreted pheromones during the conditioning period prevents a definitive quantification of the relative importance of associative learning versus feeding status (as in *Figure 3* inset with the pheromone blend). Nonetheless, these results show that *C. elegans* can alter its preference for ascarosides, highlighting the flexible role of pheromones in foraging optimization.

To provide further support that the *C. elegans* preference for pheromones can change through associative learning, we asked whether the change in preference occurs also via the association with a repellent compound, namely glycerol (*Hukema et al., 2008*). To answer this question, we performed another conditioning experiment in which young adult hermaphrodites were conditioned for 1 hr in four different scenarios deriving from all the possible combinations of ± *repellent* (glycerol) and ± *pheromone blend.* During conditioning, animals were free to dwell in a plate seeded with *E. coli* OP50, and therefore, they were always exposed to a high concentration of bacterial food. Here, uncontrolled pheromone accumulation was not an issue thanks to the short conditioning period. After conditioning, worms were tested for chemotaxis to the pheromone blend (*Figure 4A*).

We found that the preference for the pheromone blend, which is retained in the – *repellent + pheromone blend* scenario (*Figure 4B*, blue bar, $CI_{-+}=0.26 \pm 0.04$), is lost in the + *repellent + pheromone blend* condition (*Figure 4B*, yellow bar, $CI_{++} = 0.04 \pm 0.03$). Again,

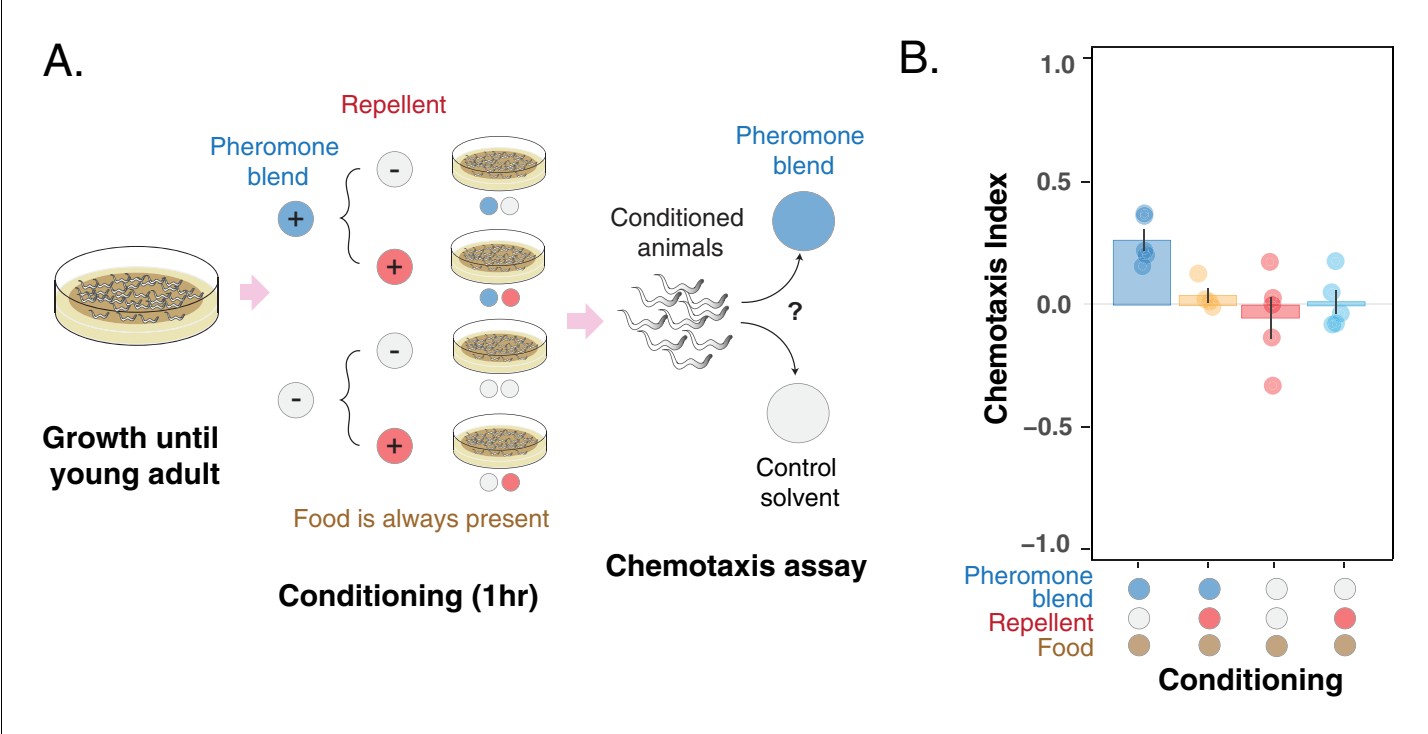

**Figure 4.** Pheromone preference changes due to association with the presence or absence of a repellent compound (glycerol). (**A**) MY1 individuals grow at high density and with plenty of food until young adult. Animals are then transferred to conditioning plates, where they spend 1 hr. Conditioning scenarios are four: + *pheromone blend – repellent*; + *pheromone blend + repellent*; – *pheromone blend + repellent*; – *pheromone blend + repellent*. Bacterial food is always abundant to prevent confounding effects due to the feeding status of animals. The short conditioning time prevents uncontrolled pheromone accumulation. Worms are then assayed for chemotaxis to the pheromone blend. (**B**) MY1 individuals are not attracted to the pheromone blend unless it is present and paired with food. Association with the repellent disrupts worm preference for pheromones gained in the presence of abundant food. Chemotaxis index is shown for the four different conditioning scenarios: + *pheromone blend – repellent* (blue bar); +*pheromone blend + repellent* (yellow bar); – *pheromone blend – repellent* (red bar); – *pheromone blend + repellent* (turquoise bar). Points indicate the outcome of each independent replicated experiments (n = 5) while bars indicate the CI ± SEM across independent experiments. The online version of this article includes the following source data for figure 4:

**Source data 1.** Conditioning with pheromones & repellent data.

when animals were conditioned in the absence of pheromones, they do not show any particular preference for the pheromone blend (CI = 0.01 ± 0.05 for + *repellent – pheromone blend* and CI = – 0.05 ± 0.08 for – *repellent – pheromone blend*, *Figure 4B* red and turquoise bars). In other words, animals do not exhibit a particular preference for pheromones, except when they are exposed to pheromones and food (in the absence of the repellent). Exposure to the repellent in the presence of food and pheromones is required to disrupt attraction. The outcome of this experiment provides further support that *C. elegans* can change its preference for the pheromone cocktail it produces through associative learning.

## Discussion

We have assessed the response of *C. elegans* to food depletion and how this influences worms' response to their pheromones. In agreement with previous studies, our findings indicate that worms exhibit different leaving times when feeding in groups on transient bacterial food patches. Interestingly, the leaving time affects *C. elegans* preference for its pheromones, with animals leaving early being attracted to their pheromones and worms leaving later being repelled by them. We showed that this inversion from attraction to repulsion depends on associative learning and appears to be an adaptive solution to optimize food intake during foraging.

Our model shows that a change from pheromone attraction to repulsion is required to optimize food intake when three different factors are combined. First, food patches that give diminishing returns, and should be abandoned when the environment provides a better average intake rate. This first factor has been thoroughly studied both theoretically and experimentally in the context of optimal foraging and the marginal value theorem (*Charnov, 1976*; *Krebs et al., 1978*; *Oaten, 1977*; *Stephens and Krebs, 1987*; *Watanabe et al., 2014*), and in this respect, our model simply reproduces previous results. A recent paper on *C. elegans* foraging contradicts this general assumption of diminishing returns, showing a near-linear depletion of food patches (*Ding et al., 2020*). This result was obtained in laboratory conditions, with very rich and uniform food patches, which may not reflect natural ones, but at least indicates that the general assumption of diminishing returns is relatively easy to break, and opens the question of whether more care should be put in assessing the conditions in which current optimal foraging models are applicable.

The second factor is competition for limited resources, which in our model creates the need to switch between pheromone-marked food patches in order to distribute the individuals more evenly. This redistribution closely resembles the Ideal Free Distribution, which postulates that animals should distribute across patches proportionally to the resources available at each source (*Bautista et al., 1995*; *Fretwell and Lucas, 1969*; *Houston and McNamara, 1987*; *Kennedy and Gray, 1993*). However, here, our model does depart from previous studies. The Ideal Free Distribution applies to cases in which the benefit per unit time decreases with the number of individuals exploiting the same resource. This is the case for habitat choice (*Fretwell and Lucas, 1969*), or if the instantaneous feeding rate decreases with the number of feeders (*Houston and McNamara, 1987*). It is not, however, the case in many foraging scenarios, including the one represented by our model (and implicitly by most optimal foraging models), in which animals can feed unimpeded by each other. In these cases, a higher number of animals simply means that the resource is depleted faster. Therefore, simply adding competition to standard optimal foraging models will not change their results qualitatively. Animals will stay in each food source until the food is so scarce that the instantaneous feeding rate falls below the environment's average. This will happen earlier for more crowded food sources, but animals will never need to switch across food sources before they are depleted.

The third key factor in our model is non-stationarity: we assume that all pheromone-marked food patches will be depleted at roughly the same time. This fact creates the need to switch before the current patch is depleted, because by then most of the benefit from undercrowded (but pheromone-marked) food patches will be gone. This non-stationary environment has received less attention than the previous factors. It is typical of species with boom-and-bust life cycles such as *C. elegans* (*Frézal and Félix, 2015*), but may also be applicable to other cases, such as migratory species (which arrive synchronously to a relatively virgin landscape), fast-dispersing invader species or, in general, species that occupy a non-stationary ecological niche.

It is interesting to note that the Evolutionary Stable Strategy found by our model does not give any benefit at the level of the species, and may even be deleterious. If we compute the average expected payoff across the whole population using the same rational as for *Equation 3*, we get $\langle H \rangle_{\text{all}} = \frac{2A_F}{N} - pc$. Therefore, increasing the probability of switching ($p$) has no effect on this population-level payoff when it's costless, and decreases it when it's costly. This Evolutionary Stable Strategy therefore emerges from intraspecific competition: Individuals benefit from paying the cost of switching to prevent being outcompeted by other individuals within the population, even if the end result is deleterious for the population as a whole.

*C. elegans* individuals use stimuli coming from the environment (smells, tastes, temperature, oxygen, and carbon dioxide levels) and from other individuals (pheromones) to efficiently navigate their habitat. An important evolutionary adaptation in this regard is that the *C. elegans* preference for each of these stimuli can change through experience, including acclimation (*Fenk and de Bono, 2017*) and associative learning phenomena (*Ardiel and Rankin, 2010*; *Colbert and Bargmann, 1995*; *Rankin, 2004*). We have identified associative learning as the most plausible phenomenon underpinning the change in pheromone preference. During feeding, worms learn to give a positive or negative preference to pheromones depending on the context in which they experience them, in particular the presence or absence of food (*Wyatt, 2014*). A similar learning process occurs in bumblebees that, in their natural habitat, do not land or probe flowers that have been recently visited and marked by chemical footprints left by themselves or other bees. It has been shown that only

experienced foragers, that is those that learnt to associate the chemical footprints with the absence of nectar in marked flowers, can successfully avoid them and increase their overall nectar intake (*Ayasse and Jarau, 2014*). This suggests that associative learning based on pairing pheromones or similar chemical signals with food availability might be frequently observed in animals feeding in groups, not only eusocial insects, as a strategy to increase food intake.

We have shown that dispersal of feeding stages of *C. elegans* from occupied patches is regulated by the recent experience of food availability and pheromones, which indicates, at any time, whether it is better to follow the scent of pheromones or to avoid it. A mechanism based on the synergistic interaction between food and pheromones also regulates *C. elegans* dispersal over longer time scales and, in general, its boom-and-bust life cycle (*Edison, 2009*; *Frézal and Félix, 2015*). Indeed, scarce food and high concentration of pheromones promote the entry into a resting stage (the dauer larva), allowing worms to survive unfavorable seasons and disperse to uncolonized rotten material, where abundant food, in turn, resumes development to adulthood (*Frézal and Félix, 2015*). Our findings establish an interesting parallel between mechanisms promoting dispersal over short and long temporal scales and highlight the important role that non-dauer stages play in exploiting transient bacterial patches. They also point to the emergence of interesting group dynamics promoted by this synergistic interaction between food and pheromones, adding to the wealth of studies addressing aggregation behaviors in *C. elegans* (*Ding et al., 2019*; *Greene et al., 2016*; *Sugi et al., 2019*).

Although we could not exclude the influence of the worms' feeding status in driving the change of preference for specific ascaroside pheromones, we showed that *C. elegans* attraction to two ascarosides can be turned into repulsion. Ascr#5 and icas#9 are potent signaling cues that are usually detected in *C. elegans* exudates (*von Reuss et al., 2012*), but other ascarosides can be abundant and the list of newly discovered compounds of the worm metabolism acting as signaling molecules is continuously expanding (*von Reuss et al., 2012*). The ability to assign a positive or negative preference to the pheromone blend through associative learning might depend also on other byproducts of worm metabolism or derive from the presence of multiple ascaroside molecules acting synergistically (*Srinivasan et al., 2012*). More studies are required to establish a link between associative learning and the composition of the pheromone blend, which is known to vary among developmental stages (*Kaplan et al., 2011*), sexes (*Izrayelit et al., 2012*) and strains (*Diaz et al., 2014*), ultimately allowing the discovery of novel roles for *C. elegans* pheromones (*Viney and Harvey, 2017*).

As a final remark, our results suggest that *C. elegans* preference for pheromones might not be innate, as it was previously stated (*Greene et al., 2016*; *Macosko et al., 2009*; *Pungaliya et al., 2009*; *Simon and Sternberg, 2002*; *Srinivasan et al., 2012*; *Srinivasan et al., 2008*) and question what it means to be a naive worm (see also *Webster and Rutz, 2020*). Worms that we call 'naive' are directly assayed for chemotaxis after being simultaneously exposed to both bacterial food and ascaroside pheromones, which are continuously excreted by the animals during their growth (*Kaplan et al., 2011*). Hence, it is possible that the attraction that 'naive' worms exhibit is due to the positive association with food that they learn to make during growth on the plate.

In conclusion, our study establishes a link between learning and social signals, providing a framework for further analysis unravelling the neuronal origin of the observed behaviors. However, the experiments presented here were performed with the natural isolate MY1. Thus, it remains to be tested if the same responses occur in the canonical lab strain N2, whose social behavior has changed due to laboratory domestication (*Sterken et al., 2015*), and in other natural strains with a social life more similar to MY1 (*Greene et al., 2016*). Nonetheless, by working with a natural isolate rather than N2, we could provide insights into the ecological significance of the inversion in the preference for pheromones and respond to the pressing need to further our knowledge of *C. elegans* ecology (*Petersen et al., 2015*).

## Materials and methods

### Strains and culture conditions

We used a *Caenorhabditis elegans* strain recently isolated from the wild, MY1 (Lingen, Germany). The strain has been obtained from the Caenorhabditis Genetic Centre (CGC). Animals were grown

at 21–23°C (room temperature) on nematode growth media (NGM) plates (100 mm) seeded with 200 µl of a saturated culture of *E. coli* OP50 bacteria (*Stiernagle, 2006*). As for OP50 culture, a single colony was inoculated into 5 ml of LB medium and grown for 24 hr at 37°C.

### Pheromones

We obtained the crude pheromone blend by growing worms in liquid culture for 9 days (at room temperature and shaking at 250 rpm) (*von Reuss et al., 2012*). Individuals from one plate were washed and added to a 1-l flask with 150 ml of S-medium inoculated with concentrated *E. coli* OP50 pellet made from 1 l of an overnight culture. Concentrated *E. coli* OP50 pellet was added any time the food supply was low, that is when the solution was no longer visibly cloudy (*Stiernagle, 2006*). The pheromone blend was then obtained by collecting the supernatant and filter-sterilizing it twice. A new pheromone blend was produced every 3 months. Pure synthetic ascarosides (ascr#5 and icas#9) were obtained from the Schroeder lab and kept at −20°C in ethanol. Each time an experiment was performed, an aqueous solution at the desired molar concentration was prepared (10 µM for ascr#5 and 10 pM for icas#9). The control solvent for the pheromone blend is S-medium, while the control solvent for the pure ascarosides is an aqueous solution with the same amount of ethanol present in the ascaroside aqueous solution (*Srinivasan et al., 2012*).

### Choice after food assay

It is a chemotaxis assay modified from *Bargmann and Horvitz, 1991* and *Saeki et al., 2001*, performed on naive worms that encounter a food patch before making the choice between the pheromone blend and the control solvent. We used 100 mm NGM plates in which we deployed 20 µl of the pheromone blend, 20 µl of control solvent and 15 µl of a diluted OP50 *E. coli* culture at equal distance from each other (*Figure 1—figure supplement 1A*). In the pheromone and control spots, 2 µl of 0.5 M sodium azide was added in order to anesthetize the animals once they reached the spots. Since the anesthetic action of sodium azide lasts for about 2 hr in this set-up, another 1 µl was added two hours after the beginning of the assay in both spots. naive animals were placed close to the bacteria spot, so that they stop and feed in the patch before chemotaxis toward the two cues. Worms are left to wander freely on the assay plate for 5 hr. The number of worms around the two spots was counted every hour and the chemotaxis index was calculated based on the number of new worms that reached the two spots during each hour.

### Chemotaxis assay

Chemotaxis assay has been performed in 60 mm NGM plates, in which worms are given the choice between pheromone (either 20 µl of the pheromone blend or 20 µl of a pure ascaroside in aqueous solution) and a control solvent (20 µl) (*Bargmann and Horvitz, 1991*; *Saeki et al., 2001*). The two spots are deployed ~3 cm apart from each other (*Figure 1—figure supplement 1B*). Shortly before the start of the assay, 1 µl of 0.5 M sodium azide is added to both spots in order to anesthetize the animals once they reach the spots. Animals, either naive or trained, are placed equidistant from the two spots and left to wander on the assay plate for 1 hr at room temperature (*Figure 1—figure supplement 1B*). The average number of worms in each experiment is indicated in the figure captions. The assay plates were then cooled at 4°C and the number of worms around each spot was counted using a lens. The chemotaxis index is then calculated as $\frac{N_p - N_c}{N_p + N_c}$, where $N_p$ is the number of worms within 1 cm of the center of the pheromone spot, while $N_c$ is the number of worms within 1 cm of the center of the control spot. The number of independent experiments (performed in different days) is indicated in each figure caption. For each experiment, we usually performed 10 replicated assays for each scenario. The average number of worms used in each replicated assay across all experiments is ~ 50.

### Conditioning experiments

Hermaphrodite individuals of the MY1 strain are grown until they become young adults in NGM plates seeded with 200 µl of saturated *E. coli* OP50 bacteria. Then, animals are washed off the plates with wash buffer (M9 + 0.1% triton), transferred to an Eppendorf tube and washed twice by spinning down the worms and replacing the supernatant with fresh wash buffer each time. After that, animals are transferred to conditioning plates. In the first series of experiments, the four different scenarios

derive from all the possible combinations of ± *food* and ± *pheromone blend*. Plates are prepared ~16 hr before the training starts, so that bacteria can grow and form a lawn. NGM plates are seeded with ±200 µl of saturated *E. coli* OP50 culture and ±200 µl of pheromone blend. Animals spend 5 hr in the conditioning plates at room temperature before being assayed for chemotaxis to the pheromone blend.

In the series of experiments with the repellent, the four different scenarios derive from all the possible combinations of ± *repellent* (glycerol) and ± *pheromone blend*. Conditioning plates are prepared ~1 before the start of the experiment and are NGM plates seeded with 200 µl of saturated *E. coli* OP50 culture ± 0.5 M glycerol and ±200 µl of pheromone blend. To keep the concentration constant, when the pheromone blend was not added, we dilute OP50 with µl of S-medium. Animals stay in the conditioning plates for one hour at room temperature before being assayed for chemotaxis to the pheromone blend.

In the experiments with pure ascarosides *ascr#5* and *icas#9*, the four different scenarios derive from all the possible combinations of ± *food* and ± *pure ascaroside* (in aqueous solution) and are prepared as the experiment with food and the pheromone blend. However, the concentration of ascaroside that was added in the conditioning plate was higher than the concentration at which the worms were tested for chemotaxis (for *ascr#5* was 10 µM, while for *icas#9* was 10 pM icas#9) to compensate for the diffusion of the ascaroside throughout the agar in the conditioning plates. Ascr#5 was added at a concentration of 100 µM onto conditioning plates, while icas#9 was added at a concentration of 1 µM. Worms spent 5 hr in the conditioning plates at room temperature, after which they are assayed for pheromone chemotaxis.

## Acknowledgements

The authors thank Sreekanth Chalasani and three anonymous reviewers for their constructive feedback on the paper; the members of the Gore lab for comments on the earlier versions of the manuscript; Ying K Zhang for assistance with the synthesis of ascarosides and Jonathan Friedman for feedback on the model. The *C. elegans* strain we used was provided by the CGC, which is funded by NIH Office of Research Infrastructure Programs (P40 OD010440). This work was supported by NIH and the Schmidt Foundation. APE was funded by EMBO Postdoctoral Fellowship Grant ALTF 818–2014, Human Frontier Science Foundation Postdoctoral Fellowship Grant LT000537/2015, and a CNRS Momentum grant.

## Additional information

### Funding

| Funder | Grant reference number | Author |
| --- | --- | --- |
| European Molecular Biology Organization | ALTF 818-2014 | Alfonso Pérez-Escudero |
| Human Frontier Science Program | LT000537/2015 | Alfonso Pérez-Escudero |
| National Institutes of Health | P40 OD010440 | Jeff Gore |
| Schmidt Family Foundation | | Jeff Gore |
| Centre National de la Recherche Scientifique | Momentum | Alfonso Pérez-Escudero |

The funders had no role in study design, data collection and interpretation, or the decision to submit the work for publication.

### Author contributions

Martina Dal Bello, Conceptualization, Data curation, Formal analysis, Visualization, Writing - original draft; Alfonso Pérez-Escudero, Conceptualization, Formal analysis, Funding acquisition, Writing - original draft; Frank C Schroeder, Resources, Writing - original draft; Jeff Gore, Conceptualization, Supervision, Funding acquisition, Writing - original draft

## Author ORCIDs
Martina Dal Bello (iD) https://orcid.org/0000-0003-3706-2929
Alfonso Pérez-Escudero (iD) https://orcid.org/0000-0002-4782-6139
Frank C Schroeder (iD) http://orcid.org/0000-0002-4420-0237
Jeff Gore (iD) https://orcid.org/0000-0003-4583-8555

## Decision letter and Author response
Decision letter https://doi.org/10.7554/eLife.58144.sa1
Author response https://doi.org/10.7554/eLife.58144.sa2

## Additional files
### Supplementary files
• Transparent reporting form

### Data availability
All data generated or analysed during this study are included in the manuscript and supporting files. Source data files have been provided for Figure 1, Figure 3, Figure 3—figure supplement 1 and Figure. 4.

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

## Appendix 1

### Foraging model

We assume that two types of food patches exist. Food patches marked with pheromones, which have a high average value (they are capable of sustaining worm growth and are easy to find). Unmarked food patches, which have a low average value as they are more difficult to find.

Initially, individuals are distributed across the pheromone-marked patches. Let $K$ be the number of patches, $n_i$ the number of individuals in the $i$-th patch (for $i = 1, 2 \ldots K$), and $N$ the total number of individuals (so $N = \sum_{i=1}^{K} n_i$). For simplicity, here we assume that all food patches are identical. We will show below that removing this assumption does not change our results qualitatively.

At any time, individuals can take three possible actions: **Remain** in the current patch, **switch** to another pheromone-marked patch (so they leave the patch and follow pheromones), or **disperse** and search for an unmarked patch (so they leave the patch and avoid pheromones).

Individuals' instantaneous feeding rate $g(t)$ depends on their choices. Let's start with the choice of dispersal. To model this decision we borrowed the results of classical foraging models from which the Marginal Value Theorem was derived (*Charnov, 1976*). These models describe an individual depleting a food patch, whose environment contains other food patches that remain stationary (i.e., on average the other food patches are not being depleted over time). Accordingly, we assume that the unmarked food patches remain stationary. In these conditions, one can compute an average expected intake rate from dispersing and searching for unmarked patches, which we will call $g_D$. This average intake rate takes into account the average quality of the unmarked food patches and the time needed to find and consume them. The optimal strategy is to remain in the current food patch until the instantaneous feeding rate ($g(t)$) falls below $g_D$ (*Charnov, 1976*). Following these models, we assume that any individual that disperses will experience a constant instantaneous feeding rate $g_D$.

While we can use the formalism of classical foraging models for the dispersal decision, we cannot do the same for the switching decision, because the pheromone-marked food patches are non-stationary (they all get depleted at roughly the same time, a feature characteristic of species with a boom-and-burst life cycle such as *C. elegans*). We will therefore model explicitly food depletion in all pheromone-marked patches.

We assume that individuals at a pheromone-marked food patch feed at a rate proportional to the amount of food left in the patch: $g(t) = A(t)/\tau$, where $A(t)$ is the amount of food available at the food patch at time $t$, and $\tau$ is a constant that determines the feeding rate. Therefore, food patches get depleted over time as $A(t) = A_0 e^{-\frac{m}{\tau}t}$, where $A_0$ is the initial food density in the pheromone-marked patches and $m$ is the number of individuals in the patch. (Proof: If $m$ worms occupy a patch, and each worm feeds at a rate $g(t) = A(t)/\tau$, then the food will be depleted at a rate $\frac{dA}{dt} = -\frac{m}{\tau}A$. Assuming that $m$ remains constant over time, the solution to this differential equation is $A(t) = A_0 e^{-\frac{m}{\tau}t}$, where $A_0$ is the initial food density.). Therefore, instantaneous feeding rate in a food patch will be

$$g(t) = \frac{A_0}{\tau} e^{-\frac{m}{\tau}t} \qquad [S1]$$

Individuals that switch pay a cost $c$ for switching. We assume that switching is fast compared to the depletion rate of the food patches, so switching is instantaneous in our model. Individuals that switch will then arrive to any pheromone-marked food patch with equal probability (including their initial one).

We now consider the total food intake, which is the integral of the instantaneous intake rate ($g(t)$) over a long period of time. The exact length of this period does not actually matter, because in all relevant cases we will be comparing strategies that end with dispersal, and therefore get the same intake rate at the end. We will always work with differences between the payoffs of these strategies, so these final periods will cancel out.

Finally, we assume that individuals have strong sensory constraints: They only perceive their instantaneous feeding rate ($g(t)$), not having information about any of the other parameters (number of patches, number of individuals per patch, etc.). However, their behavior can be adapted to be

optimal with respect to the average values of these parameters over the species' evolutionary history.

In these conditions, the following Evolutionary Stable Strategy (**Maynard Smith, 1982**; **Smith and Maynard Smith, 1974**) exists: At time $t = 0$, all individuals have a probability $p^*$ of switching (so a fraction $p^*$ of the individuals will switch). Then they all remain in the food patches until their instantaneous feeding rate falls below $g_D$, at which point they disperse. In the experiment we observed *C. elegans* individuals leaving the food patch continuously, while our model predicts that they should all switch at $t = 0$. This difference is due to the simplicity of the model, which neglects factors such as inter-individual differences and stochasticity. Despite this simplicity, the model captures the key feature that individual worms leave their initial food patch before it is depleted (switching behavior), which is a counterintuitive idea in the context of optimal foraging.

## Proof

We will prove each part of the Evolutionary Stable Strategy separately.

1. *Individuals will not disperse until the occupied food patches are nearly depleted*

Dispersing gives an instantaneous average payoff of $g_D$, so individuals should never disperse if their instantaneous intake rate is above $g_D$ (i.e., if the food density in their current patch is $A(t) > \tau g_D$). We define

$$A_D = \tau g_D, \qquad \text{[S2]}$$

which is the amount of food left in the food patch when the instantaneous feeding rate equals $g_D$.

2. *The probability to switch (p) has a stable equilibrium (p\*)*

If all individuals follow the Evolutionary Stable Strategy, at time $t = 0$ a fraction $p$ of them switches, changing the distribution of individuals across food patches. Let $m_1, m_2 \ldots m_K$ be the number of individuals in each food patch after the switch. These numbers are related to the initial distribution as

---

## Box 1. Notation index.

$A_0$ : Initial amount of food in the food patches

$A_i(t)$ : Amount of food at the $i$-th food patch at time $t$.

$A_D$ : Amount of food left in a food patch when instantaneous feeding rate matches $g_D$ ($A_D = \tau g_D$)

$A_E$ : Effective amount of food in a food patch (amount of food that worms will extract from the food patch, above what they would obtain by dispersing from the beginning).

$c$: Cost of switching

$g(t)$ : Instantaneous feeding rate

$g_D$ : Average instantaneous feeding rate after dispersal

$G$ : Total food intake.

$H$ : Payoff (total food intake minus cost)

$\langle H \rangle$ : Expected payoff

$\Delta H$ : Benefit of switching, $\Delta H = \langle H \rangle_{switch} - \langle H \rangle_{Remain}$

$K$ : Number of food patches

$m_i$ : Number of worms in the $i$-th food patch after the switch (so for any $t > 0$). In the ESS these numbers remain constant until worms start to disperse.

$n_i$ : Initial number of worms in the $i$-th food patch (before any worm switches or disperses)

$N$ : Total number of worms ($N = \sum_{i=1}^{k} n_i$)

$p$: Probability of switching at $t = 0$.

$p*$ : Value of $p$ at the ESS.

$\tau$ : Inverse of feeding rate

$$m_i = n_i(1-p) + \frac{Np}{K},$$ [S3]

which is the number of individuals that remained in the $i$-th patch plus the number of individuals that arrive to the $i$-th patch after the switch. Here we are assuming that worms that switch have equal probability to arrive to any occupied food patch (including the initial one).

After the switch, all individuals will remain in their new food patch until the instantaneous feeding rate reaches $g_D$. From **Equation S1**, this will happen at time

$$t_{D,i} = \frac{\tau \log(A_0/A_D)}{m_i}$$ [S4]

for the $i$-th food patch.

We are ready to compute the total intake over a period $T$ for an individual at the $i$-th patch. It's convenient to split this in the two periods before and after dispersal, so we have

$$G_i = \int_0^T g(t)dt = \int_0^{t_{D,i}} \frac{A_0}{\tau} e^{-\frac{m_i}{\tau}t}dt + \int_{t_{D,i}}^T g_D dt$$ [S5]

where we have used **Equation S1** for the period inside the food patch, and the constant feeding rate $g_D$ for the period after dispersal. Solving these two integrals and replacing $t_{D,i}$ for its expression in **Equation S4** gives

$$G_i = \frac{A_0 - A_D - A_D \log\frac{A_0}{A_D}}{m_i} + g_D T.$$ [S6]

We now define the effective amount of food in the food patch as

$$A_E = A_0 - A_D - A_D \log\frac{A_0}{A_D}.$$ [S7]

This effective amount of food represents the benefit that worms can extract from the food patch: $A_0 - A_D$ is the amount of food they will eat from the food patch, and $-A_D \log\frac{A_0}{A_D}$ is a correction for the amount of food they would had been eating over the same period if they had dispersed from the beginning. We then have

$$G_i = \frac{A_E}{m_i} + g_D T.$$ [S8]

Now we can compute the expected payoff for each decision ($\langle H \rangle$), which is the expected total food intake minus any costs incurred by the behavior. Individuals that switch have an equal probability of ending up in any of the food patches, so their expected payoff is simply the average of the payoffs across the food patches minus the cost of switching:

$$\langle H \rangle_{Switch} = \sum_{i=1}^K \frac{1}{K}\frac{A_E}{m_i} + g_D T - c.$$ [S9a]

In contrast, individuals that remain have a probability $n_i/N$ of being in the $i$-th patch, so their expected payoff is

$$\langle H \rangle_{Remain} = \sum_{i=1}^K \frac{n_i}{N}\frac{A_E}{m_i} + g_D T.$$ [S9b]

We now compute the benefit of switching,

$$H = \langle H \rangle_{Switch} - \langle H \rangle_{Remain} = \sum_{i=1}^K \left(\frac{1}{K} - \frac{n_i}{N}\right)\frac{A_E}{m_i} - c.$$ [S10]

Note that this benefit of switching is identical to what we would obtain using **Equations 3a and**

*3b* in the main text instead of *Equations S9a and S9b*. For this reason, this model and the simplified model presented in the main text are mathematically equivalent.

It is now convenient to define

$$n_i = n_i - N/K,$$ [S11]

which is the deviation in the initial number of individuals from the average number of individuals in every food patch. We also substitute $m_i$ according to *Equation S3*, getting

$$H = -\frac{1}{N}\sum_{i=1}^{K} n_i \frac{A_E}{n_i(1-p) + \frac{Np}{K}} - c.$$ [S12]

The equilibrium value of $p$, or $p^*$ will be such that $\Delta H = 0$, so

$$-\frac{1}{N}\sum_{i=1}^{K} n_i \frac{A_E}{n_i(1-p^*) + \frac{Np^*}{K}} - c = 0.$$ [S13]

We did not find a simple analytical expression for the value of $p^*$, but we can make several observations:

1. $\Delta n_i$ are zero, the first term of $\Delta H$ is always zero, so $\Delta H \leq 0$ and switching is never advantageous (therefore, $p^* = 0$). This makes intuitive sense: If all $\Delta n_i$ are zero, the individuals were initially distributed in the optimal way (equally distributed across the food patches), so switching cannot bring any benefit.

2. If $c = 0$, then $p^* = 1$ regardless of the value of the rest of the parameters (this can be checked by substitution in *Equation S13*). Therefore, if switching is costless all individuals should switch regardless of the values of the other parameters.

3. When $p = 1$, $\Delta H = -c$

4. When $p = 0$, $H = -\frac{A_E}{N}\sum_{i=1}^{K} \frac{n_i}{n_i} - c$

5. $\Delta H$ decreases monotonically as $p$ increases, for any values of the parameters.

(Proof: $\frac{\partial \Delta H}{\partial p} = -\frac{A_E}{N}\sum_{i=1}^{K} \frac{n_i\left(n_i - \frac{N}{K}\right)}{\left(n_i(1-p) + \frac{Np}{K}\right)^2} = -\frac{A_E}{N}\sum_{i=1}^{K} \frac{n_i^2}{\left(n_i(1-p) + \frac{Np}{K}\right)^2}$. This is always negative as long as $A_E > 0$, because $N$ is always positive and all terms inside the sum are squared.).

Therefore, $\Delta H$ decreases monotonically between $\left(-\frac{A_E}{N}\sum_{i=1}^{K} \frac{\Delta n_i}{n_i} - c\right)$ and $-c$, and the value of $p$ at which $\Delta H = 0$ the equilibrium $p^*$ of our Evolutionary Stable Strategy (*Figure 2—figure supplement 1*). If $c < -\frac{A_E}{N}\sum_{i=1}^{K} \frac{\Delta n_i}{n_i}$, then $p^*$ is greater than 0 and a fraction of the population will switch. If $c > -\frac{A_E}{N}\sum_{i=1}^{K} \frac{\Delta n_i}{n_i}$, then $p^* = 0$.

In the equilibrium, no mutant has an incentive to deviate from its strategy, since both switching and remaining give the same payoff. Furthermore, the equilibrium is stable: When $p > p^*$, $\Delta H$ becomes negative, meaning that individuals that switched get lower payoff than individuals that remained, and pushing the population toward lower $p$. Conversely, when $p < p^*$, individuals that switched have an advantage and the system is pushed toward higher $p$.

*3. Individuals must not switch more than once*

Individuals that switched at $t = 0$ have the same probability of being in every food patch. A new switch will leave these probabilities unchanged—we assume that the population is large enough so that a single mutant does not alter the distributions significantly., so will not affect the expected payoff. Therefore, individuals have no incentive to switch more than once.

*4. Individuals must switch at t=0.*

A mutant that delays the switch to some later time $t_S > 0$ will spend its time before the switch in an overcrowded patch (on average) and will therefore get lower final payoff than the wild-type that switches at time $t = 0$. Let's see it mathematically, comparing the expected payoff for switching at $t = 0$ and the expected payoff for switching at $t = t_S$:

$$\langle H\rangle_{Switchatt=0} - \langle H\rangle_{Switchatt=t_S} = \sum_{i=1}^{K}\frac{1}{K}\int_{0}^{t_S}\frac{A_0}{\tau}e^{-\frac{m_i}{\tau}t}dt + \sum_{i=1}^{K}\frac{1}{K}\int_{t_S}^{t_{D,i}}\frac{A_0}{\tau}e^{-\frac{m_i}{\tau}t}dt + \sum_{i=1}^{K}\frac{1}{K}\int_{t_{D,i}}^{T}g_D dt - c$$
$$- \sum_{i=1}^{K}\frac{n_i}{N}\int_{0}^{t_S}\frac{A_0}{\tau}e^{-\frac{m_i}{\tau}t}dt - \sum_{i=1}^{K}\frac{1}{K}\int_{t_S}^{t_{D,i}}\frac{A_0}{\tau}e^{-\frac{m_i}{\tau}t}dt - \sum_{i=1}^{K}\frac{1}{K}\int_{t_{D,i}}^{T}g_D dt + c \qquad [S14]$$

Most of the terms cancel out, leaving

$$\langle H\rangle_{Switchatt=0} - \langle H\rangle_{Switchatt=t_S} = \sum_{i=1}^{K}\left(\frac{1}{K}-\frac{n_i}{N}\right)\int_{0}^{t_S}\frac{A_0}{\tau}e^{-\frac{m_i}{\tau}t}dt = A_0\sum_{i=1}^{K}\left(\frac{1}{K}-\frac{n_i}{N}\right)\frac{1-e^{-\frac{m_i}{\tau}t_S}}{m_i}\text{, which is always positive.}$$

(Proof: $\sum_{i=1}^{K}\left(\frac{1}{K}-\frac{n_i}{n}\right)\frac{1-e^{-\frac{m_i}{\tau}t_S}}{m_i} = -\sum_{i=1}^{K}\Delta n_i \alpha_i$, where $\Delta n_i$ is defined as in Equation S11, and we define $\alpha_i = \frac{1-e^{-\frac{m_i}{\tau}t_S}}{m_i}$ We will show first that $\Delta n_i$ and $\alpha_i$ are perfectly anticorrelated (i.e. if $\Delta n_i > \Delta n_j$ , then $\alpha_i < \alpha_j$ for any $i,j$). Then, we will show that this implies that $\sum_{i=1}^{K}\Delta n_i \alpha_i$ must be negative. $\Delta \boldsymbol{n_i}$ and $\alpha_i$ are perfectly anticorrelated: Both $\Delta n_i$ and $\alpha_i$ depend on $n_i$. Let's see that their derivatives with respect to it have opposite signs: From *Equation S11*, $\frac{\partial \Delta n_i}{\partial n_i} = 1$, so it's positive. From Equation S3, $\frac{\partial m_i}{\partial n_i} = 1$, so $\frac{\partial \alpha_i}{\partial n_i}$ has the same sign as $\frac{\partial \alpha_i}{\partial m_i} = \frac{e^{-\frac{m_i}{\tau}t_S}\left(\frac{m_i}{\tau}t_S + 1 - e^{\frac{m_i}{\tau}t_S}\right)}{m_i^2}$. Given that $e^{-\frac{m_i}{\tau}t_S}$ and $m_i^2$ are always positive, this has the same sign as $\left(\frac{m_i}{\tau}t_S + 1 - e^{\frac{m_i}{\tau}t_S}\right)$, which is always negative it's zero for $m_i = 0$, and $\frac{\partial}{\partial m_i}\left(\frac{m_i}{\tau}t_S + 1 - e^{\frac{m_i}{\tau}t_S}\right) = t_S\left(1 - e^{\frac{m_i}{\tau}t_S}\right) \leq 0$ as long as $t_S \geq 0$ and $\frac{m_i}{\tau} \geq 0$, which is always true. Therefore, $\frac{\partial \alpha_i}{\partial n_i}$ is always negative.).

Therefore, switching at $t = 0$ is advantageous.

*5. Individuals must disperse once their current food patch is depleted.*

Once $g(t) = g_D$, remaining in the same patch will lead to an instantaneous feeding rate below $g_D$, because $g(t)$ decreases over time. Therefore, worms should not remain. And neither they should switch, as we saw in sections 3 and 4. Therefore, they should disperse.

## Food patches of different sizes and qualities

The above calculations assume that all food patches are identical, having an initial density $A_0$ and the same sizes. In reality, food patches will differ both in their initial density and their size.

Differences in the initial density ($A_0$) lead to effects that have been thoroughly discussed in the optimal foraging literature (*Stephens and Krebs, 1986*). Higher density translates into higher feeding rate, which can be detected by the worms as they exploit the patches. Worms that estimate that their current patch has lower quality than the environmental average should leave it and search for a more profitable one, and the key challenge is the estimation of this environmental average quality. This effect will be superimposed to the one described by our model, but we don't expect any qualitatively new effect emerging from this interaction. Part of the merit of our work is to show that even in the absence of these differences in initial density, worms may benefit from leaving a non-depleted food patch.

Differences in patch size are more interesting from our point of view. A larger food patch will not give a higher instantaneous feeding rate, but will be depleted more slowly. So the amount of food in the patch will be

$$a_i(t) = A_0 e^{-\frac{mt}{\tau s_i}}, \qquad [S15]$$

where $s_i$ is the size of the $i$-th food patch. We assume that worms cannot detect the size of their current food patch (the only way to do so would be by measuring depletion rate, which is more difficult than measuring instantaneous feeding rate and requires waiting for a long enough period of time for depletion to be significant). Therefore, the behaviors available to the worms are the same as before, and they still have an equal probability of arriving to any food patch when they decide to switch. Repeating the calculations made above but using *Equation S15* as instantaneous density, the benefit of switching becomes

$$\Delta H = -\frac{1}{N}\sum_{i=1}^{K}\Delta n_i \frac{s_i A_E}{n_i(1-p)+\frac{Np}{K}} - c, \tag{S16}$$

with an equilibrium probability given by

$$-\frac{1}{N}\sum_{i=1}^{K} n_i \frac{s_i A_E}{n_i(1-p^*)+\frac{Np^*}{K}} - c = 0. \tag{S17}$$

Unlike the case with identical food patches, even for $c = 0$ the equilibrium probability can be different than 1, and it depends on the initial distribution of individuals, and the distribution of patch sizes. In fact, it is possible to start in a situation in which switching is detrimental: If large patches contain more individuals at the beginning, remaining in the patches will produce a more balanced depletion than switching to equalize the number of individuals per patch. However, this good match between patch size and number of individuals is unlikely. In nature one would expect little or no correlation between patch size and initial number of individuals. In this case, the average result is very similar to the case with identical food patches (*Figure 2—figure supplement 2*).

