## [Decision Letter]

**Acceptance summary:**

This paper will be of interest to scientists in the field of animal behaviour, especially those working on foraging, navigation and the integration of sensory cues. Experimental data obtained with *C. elegans* convincingly demonstrate an inversion of preferences depending on the presence of other individuals. The authors interpret this behaviour as an optimal foraging strategy and suggest a model that might represent a first step towards a theoretical understanding of these observations.

**Decision letter after peer review:**

Thank you for submitting your article "Inversion of pheromone preference optimizes foraging in *C. elegans*" for consideration by *eLife*. Your article has been reviewed by three peer reviewers, and the evaluation has been overseen by a Reviewing Editor and Christian Rutz as the Senior Editor. The reviewers have opted to remain anonymous.

The reviewers have discussed their reviews with one another, and the Reviewing Editor has drafted this decision letter to help you prepare a revised submission.

Our expectation is that you will eventually carry out the additional experiments and report on how they affect the relevant conclusions either in a preprint on bioRxiv or medRxiv, or if appropriate, as a Research Advance in *eLife*, either of which would be linked to the original paper.

Summary:

The authors present an analysis of the foraging dynamics of *C. elegans* in patchy food environments and show how they bias their decisions depending on feeding history and sensory cues that reflect the density of worms. Navigational preference for these sensory cues is found to change from attractive to repulsive depending on the time at which worms leave a food patch, and additional experiments that condition worms under different combinations of conditions indicate that associative learning is involved in this inversion of preference. This behavior is illuminated by a mathematical model that points to the conclusion that this inversion represents an optimal, evolutionarily stable foraging strategy.

Essential revisions:

Overall, the results are interesting and clearly presented but could be better supported by experiments and modeling. The paper is of high quality and would be of interest to the readership of *eLife* provided that the following crucial remarks on experiments and modeling are duly addressed by a major revision.

Experiments:

Regarding the interpretation of the data in Figure S2, which was designed to test for associative learning in the preference for specific pheromone molecules (synthetic ascarosides). You state in the main text that the results in Figure 3 of associative learning experiments with the 'pheromone blend' (i.e., supernatant from worms in liquid culture) could be 'recapitulated' with these pure forms of the pheromone molecules, but the data appear to contradict this statement. It appears that difference in navigational preference for these pheromone molecules exist between worms conditioned in the +food and -food conditions, regardless of whether the pheromone molecules are present during the conditioning. Thus, it appears as if the preference for these pheromones might be explained by feeding status only, without associative learning. Have we missed something here? The results of Figure 3 are still consistent with associative learning, but the results of Figure S2 appear to suggest that the cues involved in the associative learning are different from the specific pheromones tested within Figure S2, and must in fact override the feeding-state dependent preference to these specific ascaroside molecules. This leaves us wondering whether it is fair to conclude that the sensory cues involved in the associative learning are really pheromone molecules. Can you rule out that the learning is a response to other substances in the supernatant, e.g. (in their own words) "other products of worm metabolism, compounds deriving from the decomposition of dead worms and bacteria, and perhaps other unknown substances"?

You may want to consider conducting additional experiments (to find other pheromone molecules that do demonstrate evidence for associative learning), or to just change relevant wording to avoid committing to what appears to be at present a weakly supported conclusion.

Mathematical modeling:

Although conceptually appealing, the modeling appears to be an add-on to the experimental results. It also seems to suffer from weaknesses that should be addressed as detailed below. First, it would seem that at least some of the analysis results (e.g., that equal petitioning would be optimal) might depend on the assumption that all patches are of equal size. This does seem a strong assumption if the model is to be considered relevant to natural ecology. Which of the model-based conclusions are still valid if this assumption is removed? Second, how reasonable is the assumption that all worms start in all patches at the same time? Naively, it would seem more realistic that a small founder population would arrive at a patch, and that these foraging dynamics would play out in a context that involves not only food depletion but also population growth. Would the main conclusions of the model (about optimal foraging and evolutionary stability) still hold in a more realistic model that considers such expected natural population dynamics? Third, the key conceptual assumption is that all patches are either colonized together or left unoccupied. It would be helpful to provide references supporting this assumption. Fourth, the statement at line 135 that "worms have no way to tell if they are in the overcrowded or under-crowded environment" seems at first sight to be inconsistent with the very idea of pheromone signalling. Can you please clarify? Since the mathematical model -- currently relegated to the supplement -- forms an important part of the manuscript, we think that it should be clearly summarized in the main text with an accompanying figure, taking into account all the concerns raised above.

- Why was the natural isolate MY1 used? Would the results change with a different strain? Would it be feasible to consider a mutant with blocked pheromone sensing as a null control?

- Could the observed behavior be related to recent observations of nontrivial collective behavior in *C. elegans* (e.g., "*C. elegans* collectively forms dynamical networks", Sugi et al., 2019, Nat. Comm.; "Shared behavioral mechanisms underlie *C. elegans* aggregation and swarming", Ding et al., 2019, *eLife*)?

- The worms are described in the text to be constantly leaving the patch but the supplement states that each worm only leaves when the food intake decreases below a certain threshold value. Could you please explain the seemingly contradictory statements?

- You might want to consider citing "Information socialtaxis and efficient collective behavior emerging in groups of information-seeking agents" by Karpas et al., (2017, PNAS), and related work, on how optimal foraging is different in group as opposed to individual contexts.

- You might also want to consider the following reference, perhaps relevant for the discussion: "Measuring *Caenorhabditis elegans* Spatial Foraging and Food Intake Using Bioluminescent Bacteria" (Ding et al., 2020, Genetics).

[Editors' note: further revisions were suggested prior to acceptance, as described below.]

Thank you for submitting your article "Inversion of pheromone preference optimizes foraging in *C. elegans*" for consideration by *eLife*. Your article has been reviewed by three peer reviewers, and the evaluation has been overseen by a Reviewing Editor and Christian Rutz as the Senior Editor. The following individual involved in the review of your submission has agreed to reveal their identity: Sreekanth H Chalasani (Reviewer #4).

The reviewers have discussed their reviews with one another, and the Reviewing Editor has drafted this decision letter to help you prepare a revised submission.

Summary:

The reviewers concur that this article offers an interesting conclusion regarding optimal foraging and chemosensory valence. However, they also agree that it would benefit from a second round of revision, aiming at an improved precision of language and a better discussion of the assumptions of the model and experimental conclusions.

Essential Revisions:

1) A summary of the model and its most important parameters appears to be necessary. The revised version is still unsatisfactory from this point of view.

2) The paper still requires some improvement on the precision of language.

3) The discussion of the assumptions of the model and experimental conclusions could be further improved.

For more detail about these points, please read the Recommendations by the Reviewers.

*Reviewer #1:*

Many aspects of the manuscript have improved in this revision but unfortunately, I feel that the authors still haven't adequately incorporated the model details into the manuscript. I do feel strongly that one shouldn't show model predictions (Figure 2) without a summary of the model and its most important parameters. Such a summary is necessary so that readers can reasonable judge the modeling choices. I also disagree with the authors own assessment "The model is now extensively described in the main text (lines 123-159)". One cannot read these lines and easily recreate any sense of the model. There are many more details given in the "Foraging model" section of the supplement, but these details are also not presented in such a way so that the model architecture is clear.

One small comment for the abstract:

Line 22 consider removing "model" or changing to "animal model" to distinguish from the computational model

*Reviewer #2:*

The authors present experiments that demonstrate how *C. elegans* worms bias their foraging decisions depending on feeding history and sensory cues (here, called pheromones) that reflect the density of worms. Navigational preference for these sensory cues is found to change from attractive to repulsive depending on the time at which worms leave a food patch, and additional experiments that condition worms under different combinations of conditions (with/without the sensory cues, with/without food, with/without repellent) indicate that associative learning is involved in this inversion of preference. A mathematical model is provided to argue that this inversion represents an optimal foraging strategy that is also evolutionarily stable.

I am satisfied that the edits made by the authors sufficiently addressed concerns I raised in the previous round.

*Reviewer #4:*

The authors use the nematode *C. elegans* to reveal how animals associate social signals with specific contexts and modify their behaviors. Specifically, they show that *C. elegans* leaving a food patch are attracted to pheromonal cues, while those leaving later are repelled from pheromones. The authors using a behavioral model to suggest that the switch from attraction to repulsion is likely due to a change in learning. This study links learning with social signals providing a framework for further analysis into the underlying neuronal pathways.

Line 36 + 46: The word decision is used but lacks definition. Decision-making in *C. elegans* is controversial and should require precise language in using. If there is a decision to be made regarding exploitation of a food patch, what are the possible actions that the animal can choose to take?

Line 58: Similarly, the term associative learning requires definition.

Line 99: As stated in the rebuttal, MY1 was used as (1) it was deemed to be more natural/ethological and (2) it is known to respond to synthetic pheromones. It is unclear whether or not the use of N2 in these experiments would change the results. If these results do not hold in N2, this is of note because it could lead to interesting follow-up experiments aiming to identify the biology underlying the MY1-specific behavior. The use of MY1 is cause for concern in truly placing these results within the full body of *C. elegans* literature. It seems prudent that the use of this strain be further addressed in the discussion.

How far do pheromones diffuse to (within the detectable limit of a *C. elegans*)? Is it clear that the decision to leave a patch is made without knowledge of other food patches? Or is it possible that the animals are receiving pheromonal cues about other patches even while residing on a food patch bathed in pheromones. This distinction seems important to both the conclusions of the paper and the model and are not discussed in the body of the text. Can the diffusion of pheromones be experimentally defined or modeled to support the assumptions of the foraging model?

Is this foraging model ethological? Would worms not have a much lower payoff of leaving their patch in their natural boom-and-bust environment? Specifically, would patch density not be significantly lower in a natural environment such that switching patches is not beneficial until much later in food depletion (which could presumably be generations later)? Consider the spatial scale of a *C. elegans* to a rotting fruit.

Line 218: This brings up an interesting discussion point. The model and experiments assume stationarity of the pheromone blend over time with an inversion of valence occurring due to associative learning and satiety. However, would the pheromone blend not change throughout the course of the experiment? Could the specific combination of pheromones in the blend possibly be cause for the valence change?

Overall, I think this paper offers an interesting conclusion regarding optimal foraging and chemosensory valence. However, I do think it would benefit from precision of language and better discussion of the assumptions of the model and experimental conclusions.

[Editors' note: further revisions were suggested prior to acceptance, as described below.]

Thank you for resubmitting your work entitled "Inversion of pheromone preference optimizes foraging in *C. elegans*" for further consideration by *eLife*. Your revised article has been evaluated by Christian Rutz (Senior Editor), Antonio Celani (Reviewing Editor), and two reviewers.

Summary:

This paper will be of interest to scientists in the field of animal behaviour with a focus on foraging, navigation and the integration of sensory cues. Experimental data obtained with *C. elegans* convincingly demonstrate an inversion of preferences depending on the presence of other individuals. The authors interpret this behaviour as an optimal foraging strategy but the modelling in support of this conclusion could be improved.

Del Bello et al., present experiments that demonstrate how *C. elegans* worms bias their foraging decisions depending on feeding history and sensory cues (pheromones) that reflect the density of worms. Navigational preference for these sensory cues is found to change from attractive to repulsive depending on the time at which worms leave a food patch, and additional experiments that condition worms under different combinations of conditions (with/without the sensory cues, with/without food, with/without repellent) indicate that associative learning is involved in this inversion of preference. A mathematical model is provided to argue that this inversion represents an optimal foraging strategy that is also evolutionarily stable.

The paper offers an interesting conclusion regarding optimal foraging and chemosensory valence that is mostly supported by the data. However, the modelling part is comparatively weaker. Below is a list of the issues that still need to be addressed:

– How reasonable is the assumption that all worms start in all patches at the same time? Naively, it would seem more realistic that a small founder population would arrive at a patch, and that these foraging dynamics would play out in a context that involves not only food depletion but also population growth. Would the main conclusions of the model (about optimal foraging and evolutionary stability) still hold in a more realistic model that considers such expected natural population dynamics?

– How far do pheromones diffuse to (within the detectable limit of a *C. elegans*)? Is it clear that the decision to leave a patch is made without knowledge of other food patches? Or is it possible that the animals are receiving pheromonal cues about other patches even while residing on a food patch bathed in pheromones. This distinction seems important to both the conclusions of the paper and the model and is not discussed in the main text. Can the diffusion of pheromones be experimentally defined or modeled, to support the assumptions of the foraging model?

– Is this foraging model ethological? Would worms not have a much lower payoff of leaving their patch in their natural boom-and-bust environment? Specifically, would patch density not be significantly lower in a natural environment such that switching patches is not beneficial until much later in food depletion (which could presumably be generations later)?

– The model and experiments assume stationarity of the pheromone blend over time with an inversion of valence occurring due to associative learning and satiety. However, would the pheromone blend not change throughout the course of the experiment? Could the specific combination of pheromones in the blend possibly be cause for the valence change?

– Finally, we wish to draw your attention to the STRANGE framework for animal behaviour research, which is currently being adopted by *eLife* and seems relevant to your study: https://doi.org/10.1038/d41586-020-01751-5

*Reviewer #1:*

I am now satisfied with the improved presentation of model details.

I do note a typo in the caption of Figure 2: "During the first phase, worms equalize occupancy the occupied patches."

*Reviewer #4:*

Gore et al., show that *C. elegans* leaving food patches at different times and different preferences to pheromones. Animals leaving early are attracted, while those leaving later are repelled from pheromones. I have no concerns with the data. Most of my concerns are with the model.

It remains unclear whether the inversion is due the association of pheromones with food odorants or due to food-deprivation. I would recommend either testing these two possibilities or clarifying this in the manuscript with specific predictions for each outcome.

Line 192 – Evolutionary Stable Strategy is introduced for the first time, but is neither defined nor cited. Further, small typo exists with the word "the" on this line.

---

## [Author Response]

Essential revisions:Overall, the results are interesting and clearly presented but could be better supported by experiments and modeling. The paper is of high quality and would be of interest to the readership of eLife provided that the following crucial remarks on experiments and modeling are duly addressed by a major revision.Experiments:Regarding the interpretation of the data in Figure S2, which was designed to test for associative learning in the preference for specific pheromone molecules (synthetic ascarosides). You state in the main text that the results in Figure 3 of associative learning experiments with the 'pheromone blend' (i.e., supernatant from worms in liquid culture) could be 'recapitulated' with these pure forms of the pheromone molecules, but the data appear to contradict this statement. It appears that difference in navigational preference for these pheromone molecules exist between worms conditioned in the +food and -food conditions, regardless of whether the pheromone molecules are present during the conditioning. Thus, it appears as if the preference for these pheromones might be explained by feeding status only, without associative learning. Have we missed something here? The results of Figure 3 are still consistent with associative learning, but the results of Figure S2 appear to suggest that the cues involved in the associative learning are different from the specific pheromones tested within Figure S2, and must in fact override the feeding-state dependent preference to these specific ascaroside molecules. This leaves us wondering whether it is fair to conclude that the sensory cues involved in the associative learning are really pheromone molecules. Can you rule out that the learning is a response to other substances in the supernatant, e.g. (in their own words) "other products of worm metabolism, compounds deriving from the decomposition of dead worms and bacteria, and perhaps other unknown substances"?

We apologize that we were not more clear regarding the similarities and differences between the experiments with the two synthetic ascarosides (ascr#5 and icas#9, Figure S2 – now Figure 3 —figure supplement 1) as compared to those done with the pheromone blend (Figure 3). As we now specify in our revised version of the main text (see below), the experiments with the synthetic ascarosides were performed at high worm density for practical reasons and their results mimic those with the pheromone blend done at high worm density (inset Figure 3B). Animals conditioned without ascarosides (with and without food) could be exposed to the cocktail of pheromones excreted during the conditioning period. This is a confounding factor that prevented us from collecting definitive evidence that associative learning was the only mechanism driving the change in preference for both ascarosides. Nonetheless, our experiments do indicate that *C. elegans* can change its preference for two of its pheromones, demonstrating that these ascarosides are among the signaling molecules involved in the optimization of foraging behavior. We have therefore replaced the previous discussion with the following paragraph to the main text (lines 220-236):

“In addition to a cocktail of ascaroside pheromones, the pheromone blend contains other products of worm metabolism, compounds deriving from the decomposition of dead worms and bacteria, and perhaps other unknown substances. Worms could in principle learn and change preference for any of these compounds and thus forage efficiently as indicated by our model. To probe whether specific pheromones are involved in the foraging optimization we asked two questions: 1. Can *C. elegans* attraction to ascarosides be turned into repulsion? 2. Can *C. elegans* learn with ascarosides? To address these questions, we conducted the conditioning experiments with two synthetic ascarosides, ascr#5 and icas#9, instead of the pheromone blend. We found that *C. elegans* can change its preference for both ascarosides (Figure 3 —figure supplement 1), suggesting that ascaroside pheromones are likely contributing to the response to the pheromone blend found previously. However, these experiments were performed at high worm density, meaning that accumulation of secreted pheromones during the conditioning period prevents a definitive quantification of the relative importance of associative learning versus feeding status (as in Figure 3 inset with the pheromone blend). Nonetheless, these results show that *C. elegans* can alter its preference for ascarosides, highlighting the flexible role of pheromones in foraging optimization.”

In addition, we have changed the caption of Figure 3 —figure supplement 1 (previous Figure S2), which now reads:

“Attraction towards two ascarosides, ascr#5 and icas#9, can be turned into repulsion. A. Schematic of the conditioning experiment. Worms grow at high density and with plenty of food until they are young adults. Animals are then transferred to conditioning plates, where they spend 5 hours, before being assayed for chemotaxis to the pure ascaroside. B. Chemotaxis index (CI) is shown for the four different conditioning scenarios: + food + ascaroside (blue bars); – food + ascaroside (yellow bars); + food– ascaroside (red bars); – food – ascaroside (turquoise bars). Points indicate the outcome of each independent replicated experiment (n=4 for each ascaroside) while bars indicate the CI ± SEM across independent experiments. If attraction is turned into repulsion by the feeding status only, we expect the blue and red bars not to differ and the CI to be positive, while the yellow and turquoise bars not to differ and the CI to be either negative or equal to zero. In the experiment with ascr#5 (on the left), the CI is positive in the + food + ascaroside and the + food– ascaroside scenarios, but it differs in the two conditions without food, being negative in the – food + ascaroside scenario and not differing from zero in the – food – ascaroside scenario. These results are consistent with a combination of associative learning and feeding status driving the change in preference for ascr#5. In the experiment with icas#9 (on the right), the CI in the two conditions with food does not differ, mostly because of the large variability among the repetitions of the + food– ascaroside scenario (red points). The CI in the two conditions without food is instead negative. The most parsimonious explanation for the change in preference for icas#9 is the different feeding status rather than associative learning, although the accumulation of secreted pheromones during the conditioning period (due to the high worm density in the conditioning plates) may once again obscure the evidence of associative learning (as in the inset of Figure 3).”

We have also added a paragraph in the discussion (lines 370-382):

“Although we could not exclude the influence of the worms’ feeding status in driving the change of preference for specific ascaroside pheromones, we showed that *C. elegans* attraction to two ascarosides can be turned into repulsion. Ascr#5 and icas#9 are potent signaling cues that are usually detected in *C. elegans* exudates (Von Reuss et al., 2012), but other ascarosides can be abundant and the list of newly discovered compounds of the worm metabolism acting as signaling molecules is continuously expanding (Von Reuss et al., 2012). The ability to assign a positive or negative preference to the pheromone blend through associative learning might depend also on other byproducts of worm metabolism or derive from the presence of multiple ascaroside molecules acting synergistically (Srinivasan et al., 2012). More studies are required to establish a link between associative learning and the composition of the pheromone blend, which is known to vary among developmental stages (Kaplan et al., 2011), sexes (Izrayelit et al., 2012) and strains (Diaz et al., 2014), ultimately allowing the discovery of novel roles for *C. elegans* pheromones (Viney and Harvey, 2017).”

You may want to consider conducting additional experiments (to find other pheromone molecules that do demonstrate evidence for associative learning), or to just change relevant wording to avoid committing to what appears to be at present a weakly supported conclusion.

We agree that experiments performed at low worm density or experiments using different ascarosides could help strengthen our findings. However, given the current situation, we decided to change our wording and tone down the conclusion that particular pheromones are the only molecules involved in the observed associative learning (lines 220-236). We also added a paragraph in the discussion detailing why the complex cocktail of pheromones is, as a whole, the most plausible sensory cue driving associative learning, although more studies would be required to establish the link between the composition of the pheromone blend and associative learning phenomena (lines 370-382). See previous point.

Mathematical modeling:Although conceptually appealing, the modeling appears to be an add-on to the experimental results. It also seems to suffer from weaknesses that should be addressed as detailed below. First, it would seem that at least some of the analysis results (e.g., that equal petitioning would be optimal) might depend on the assumption that all patches are of equal size. This does seem a strong assumption if the model is to be considered relevant to natural ecology. Which of the model-based conclusions are still valid if this assumption is removed?

We agree with the reviewers that, under natural conditions, food patches can exhibit different initial bacterial density and/or size. Differences in the initial density of food have been thoroughly discussed in the optimal foraging literature and are usually reflected by differences in feeding rates. We don’t expect any qualitatively new effect emerging from this. In this revision, we focused on differences in patch size, which, instead, translate into changes in the time that it takes to deplete them. By adjusting the instantaneous feeding rate to reflect this, we could show that introducing food patches with different sizes does not qualitatively change our results. Behaviors available to worms (remaining in the current patch, switching to another pheromone-marked patch or dispersing in search for unmarked patches) are still the same. While it is true that remaining in the patches is better than switching in the case in which the largest patches contain larger numbers of individuals, on average, it is still optimal to switch before the patch is depleted following pheromone cues. We discuss the details of this important addition to the model in the Supplementary material (lines 808-855), providing the changes introduced in the equations and showing the results in an additional figure (Figure 2 —figure supplement 2):

“Food patches of different sizes and qualities

The above calculations assume that all food patches are identical, having an initial density A0 and the same sizes. In reality, food patches will differ both in their initial density and their size.

[…]

Unlike the case with identical food patches, even for c=0 the equilibrium probability can be different than 1, and it depends on the initial distribution of individuals, and the distribution of patch sizes. In fact, it is possible to start in a situation in which switching is detrimental: If large patches contain more individuals at the beginning, remaining in the patches will produce a more balanced depletion than switching to equalize the number of individuals per patch. However, this good match between patch size and number of individuals is unlikely. In nature one would expect little or no correlation between patch size and initial number of individuals. In this case, the average result is very similar to the case with identical food patches (Figure 2—figure supplement 2).”

We specify in the main text that incorporating different patch sizes does not change results at lines 145-147, which now reads:

“This result is independent of any other parameters, such as number of patches, their size or initial distribution of worms (see Supplement).”

Second, how reasonable is the assumption that all worms start in all patches at the same time? Naively, it would seem more realistic that a small founder population would arrive at a patch, and that these foraging dynamics would play out in a context that involves not only food depletion but also population growth. Would the main conclusions of the model (about optimal foraging and evolutionary stability) still hold in a more realistic model that considers such expected natural population dynamics?

It would indeed be very interesting to study the effects of population growth, but we believe it is not a basic feature of our setting and would add unnecessary complication in our model. Indeed, patch-leaving in *C. elegans*, both in our experiments and also in its habitat, occurs in a timescale of a few hours, which is insufficient for significant population growth.

Third, the key conceptual assumption is that all patches are either colonized together or left unoccupied. It would be helpful to provide references supporting this assumption.

This was unclear in our previous version, as it is not strictly required that all food patches are colonized at the same time. The only requirement of our model is that all food patches are depleted at roughly the same time, which is a characteristic of boom-and-burst life cycles. The starting time of our model could be understood as the time at which the worms colonize the food patches, but it can also be understood as any other later time in which the distribution of individuals is unequal across the food patches, and this is compatible with a progressive colonization. We have eliminated the part of the sentence in the discussion that previously stated that the food patches were colonized at the same time (lines 328-329), which now reads:

“The third key factor in our model is non-stationarity: we assume that all pheromone-marked food patches will be depleted at roughly the same time.”

Fourth, the statement at line 135 that "worms have no way to tell if they are in the overcrowded or under-crowded environment" seems at first sight to be inconsistent with the very idea of pheromone signalling. Can you please clarify?

We agree with the reviewers that sentence was unclear and inconsistent with the idea of pheromone signaling. What we meant is that even if an individual can assess the density of its conspecifics (using pheromones) in its current food patch, it has no reliable information about the colonization status of other patches. As such, it cannot tell whether it is in the more crowded or less crowded patch. We have corrected the statement in the main text (lines 137-139), which now reads:

“Worms have no way to tell whether they are in the overcrowded or in the undercrowded patch because, even if they can assess worm density in their current food patch, they have no reliable information about the other.”

Since the mathematical model -- currently relegated to the supplement -- forms an important part of the manuscript, we think that it should be clearly summarized in the main text with an accompanying figure, taking into account all the concerns raised above.

We expanded the current Figure 2, which was already about the model results, adding a panel showing the expected payoff for each strategy and the equilibrium switching probabilities. The model is now extensively described in the main text (lines 123-159). Equations, proofs and details about the different conditions under examination are instead in the Supplement.

Here is how it reads in the new version of the main text:

“The scenario described in our model implies that there are two types of food patches, those that are occupied by worms and those that are not. We assumed that the occupied patches are easier to find as a result of either being nearby or because of the accumulation of pheromones secreted by the worms. Dispersing away from the occupied food patches in search for unoccupied ones therefore gives a low average payoff (gD), not being advantageous until the occupied food patches are depleted (see Supplement). However, these patches will generically be occupied in a non-equal manner, meaning that one patch will be consumed before the other. Under these conditions, we asked which strategy maximizes individual food intake. For the sake of illustration, in Figure 2 we depict the simplest scenario, with two patches that are colonized by worms and another that is uncolonized.

Initially, worms should stay in the occupied patches, but they can either remain in their initial one or switch to the other occupied patch (by leaving their current patch and following pheromones to the neighboring patch) (Figure 2A). If all worms remain in their initial patch the overcrowded food patch will be depleted faster, so worms occupying it will benefit from switching to another occupied patch (Figure 2B). Worms have no way to tell whether they are in the overcrowded or in the undercrowded patch because, even if they can assess worm density in their current food patch, they have no reliable information about the other. However, since the majority of individuals are in the overcrowded patch, every individual has a higher probability of being there. Therefore, all worms have an incentive to switch to the other occupied patch during the initial phase (blue area in Figure 2B).

How many worms should then switch? If we assume that worms attempting to switch may in fact end up in any of the occupied patches (including the initial one) with equal probability, then all worms should attempt to switch (Figure 2C, blue). This result is independent of any other parameters, such as number of patches, their size or initial distribution of worms (see Supplement). If we assume that switching to other occupied patches is costly, then the optimal switching probability will be lower (Figure 2C, orange), but the predictions of the model remain qualitatively unaltered (see Supplement). This model therefore predicts an Evolutionary Stable Strategy in which some worms leave a patch before it is depleted and follow the pheromone cue (Figure 2A). This initial phase helps equalize worm occupancy and feeding across food patches.

Once worm numbers are equalized in the two easy-to-find food patches, worms feed until the food becomes scarce. At this point, worms benefit from leaving the depleted patches (gray area in Figure 2B) and avoiding pheromones, since pheromones now mark depleted food patches. The inversion of pheromone preference therefore helps worms to disperse to unoccupied food patches. This simple model thus predicts both different leaving times and the inversion of pheromone preference and highlights that, together, these phenomena might maximize food intake of worms foraging in a patchy environment.”

- Why was the natural isolate MY1 used? Would the results change with a different strain? Would it be feasible to consider a mutant with blocked pheromone sensing as a null control?

We chose a natural isolate because we wanted to assess behavioral patterns that could be as close as possible to those exhibited by *C. elegans* in its natural habitat. We now specify this at lines 97-99, which read:

“We used young adult hermaphrodites of the natural isolate MY1 (Lingen, Germany) to assess behavioral patterns that could be as close as possible to those exhibited by *C. elegans* in its natural habitat.”

In addition, a recent study found that MY1, compared to other natural isolates, responds to synthetic pheromones (Greene et al., 2016). In the initial phases of this project, we explored changes in the behavioral response to the pheromone blend in the standard lab strain N2 — which mimicked those of MY1 at high worm density — and in a daf-22 mutant. However, we later focused just on MY1, which is the only strain for which we have a detailed analysis of foraging and changes in the behavioral responses to pheromones.

- Could the observed behavior be related to recent observations of nontrivial collective behavior in *C. elegans* (e.g., "*C. elegans* collectively forms dynamical networks", Sugi et al., 2019, Nat. Comm.; "Shared behavioral mechanisms underlie *C. elegans* aggregation and swarming", Ding et al., 2019, eLife)?

We don’t think it is strongly related because the aggregation and swarming phenomena described by the above-mentioned papers seem to require longer timescales compared to those examined in our experiments and have been studied on homogeneous bacterial lawns. In a similar setting, however, it has been shown that some ascaroside pheromones are able to promote aggregation in *C. elegans* (e.g. Greene et al., 2016 and data collected by the authors). Despite our belief that these phenomena don’t seem to apply to our results, we mentioned the potential link between our results and aggregation behavior in the discussion referring to these relevant studies (lines 363-369).

“Our findings establish an interesting parallel between mechanisms promoting dispersal over short and long temporal scales and highlight the important role that non-dauer stages play in exploiting transient bacterial patches. They also point to the emergence of interesting group dynamics promoted by this synergistic interaction between food and pheromones, adding to the wealth of studies addressing aggregation behaviors in *C. elegans* (Ding et al., 2019; Greene et al., 2016; Sugi et al., 2019).”

- The worms are described in the text to be constantly leaving the patch but the supplement states that each worm only leaves when the food intake decreases below a certain threshold value. Could you please explain the seemingly contradictory statements?

We have added a clarifying statement in the Supplement (lines 723-728) which reads:

“In the experiment we observed *C. elegans* individuals leaving the food patch continuously, while our model predicts that they should all switch at t=0. This difference is due to the simplicity of the model, which neglects factors such as inter-individual differences and stochasticity. Despite this simplicity, the model captures the key feature that individual worms leave their initial food patch before it is depleted (switching behavior), which is a counterintuitive idea in the context of optimal foraging.”

- You might want to consider citing "Information socialtaxis and efficient collective behavior emerging in groups of information-seeking agents" by Karpas et al., (2017, PNAS), and related work, on how optimal foraging is different in group as opposed to individual contexts.

Thanks for the suggestion. We have integrated this reference and others about collective foraging in the introduction.

- You might also want to consider the following reference, perhaps relevant for the discussion: "Measuring *Caenorhabditis elegans* Spatial Foraging and Food Intake Using Bioluminescent Bacteria" (Ding et al., 2020, Genetics).

We thank the reviewers for the suggestion. We added a sentence to discuss the findings of this relevant paper in the discussion (lines 303-309).

“A recent paper on *C. elegans* foraging contradicts this general assumption of diminishing returns, showing a near-linear depletion of food patches (Ding et al., 2020). This result was obtained in laboratory conditions, with very rich and uniform food patches, which may not reflect natural ones, but at least indicates that the general assumption of diminishing returns is relatively easy to break, and opens the question of whether more care should be put in assessing the conditions in which current optimal foraging models are applicable.”

[Editors' note: further revisions were suggested prior to acceptance, as described below.]

Essential Revisions:1) A summary of the model and its most important parameters appears to be necessary. The revised version is still unsatisfactory from this point of view.2) The paper still requires some improvement on the precision of language.3) The discussion of the assumptions of the model and experimental conclusions could be further improved.

We thank the Editor and Reviewers for the positive evaluation as well as for the suggestions and the summary of requested changes. Please see below for how we have modified the manuscript to address each of the detailed comments.

For more detail about these points, please read the Recommendations by the Reviewers.Reviewer #1:Many aspects of the manuscript have improved in this revision but unfortunately, I feel that the authors still haven't adequately incorporated the model details into the manuscript. I do feel strongly that one shouldn't show model predictions (Figure 2) without a summary of the model and its most important parameters. Such a summary is necessary so that readers can reasonable judge the modeling choices. I also disagree with the authors own assessment "The model is now extensively described in the main text (lines 123-159)". One cannot read these lines and easily recreate any sense of the model. There are many more details given in the "Foraging model" section of the supplement, but these details are also not presented in such a way so that the model architecture is clear.

We understand the concern of the reviewer and we further expanded the description of the model implementation (lines 122-192). To improve readability, in the main text we describe a simpler model with three food patches (two unevenly occupied and one unoccupied by worms) and provide a more complex version with n food patches and patches of different sizes in the supplement. We now detail the assumptions about the actions available to worms (switching to another occupied food patch or remaining in the current one) and include the derivation of the equations for the calculation of the expected payoff associated with both actions (Equation 3a and 3b), based on the per capita food intake achieved in each food patch (Equation 2) and the probability of reaching either of the two patches after switching (Equation 1). Finally, we describe how the model gives an evolutionary stable strategy. We have also improved Figure 2. Panel B highlights how the expected payoff for each strategy changes with the proportion of worms deciding to switch. Panel C is a heatmap showing the dependence of the equilibrium probability to switch (p*) on the cost of switching and the initial fraction of worms in the more crowded patch. We think that these improvements will allow the reader to get a concrete sense of the model.

The main text now reads (lines 122-192 and Figure 2):

“A simple model shows that the inversion of the preference for pheromones can be a strategy that optimizes foraging

Inside rotting fruits and stems where *C. elegans* forage, bacterial food is patchily distributed (Frézal and Félix, 2015; Schulenburg and Félix, 2017). We might then expect that the timing of dispersal from existing food patches and the strategies that optimize food intake are crucial for worm survival. A natural question then arises: can the behaviors we observed in our experiments provide any benefit to *C. elegans* foraging? We addressed this question with a simple theoretical model exploring the optimality of the inversion in pheromone preference in the context of foraging in a heterogeneous environment. […] Our model recapitulates the two key experimental observations highlighted in Figure 1: First, a fraction of worms will switch at the beginning, leaving the food patch before it is depleted and following pheromones to reach another occupied food patch. Second, once the food patches are depleted all worms will disperse, avoiding depleted food patches by reversing their preference for pheromones (which now mark depleted food patches). Here we have illustrated these results with a simplified model, with two identical food patches and in which worms can only switch or disperse at particular times. A more general model in which individuals can move at any time between any number of food patches of equal or different sizes gives the same the Evolutionary Stable Strategy (see Supplement).”

We have changed the notation in the supplement to be consistent with the equations included in the main text. Finally, we have added a paragraph in the discussion about the Evolutionary Stable Strategy found by our model (lines 366-374).

“It’s interesting to note that the Evolutionary Stable Strategy found by our model does not give any benefit at the level of the species, and may even be deleterious. If we compute the average expected payoff across the whole population using the same rational as for Equation 3, we get ⟨H⟩all=2AEN−pc. Therefore, increasing the probability of switching (p) has no effect on this population-level payoff when it’s costless, and decreases it when it’s costly. This Evolutionary Stable Strategy therefore emerges from intra-specific competition: Individuals benefit from paying the cost of switching to prevent being outcompeted by other individuals within the population, even if the end result is deleterious for the population as a whole.”

One small comment for the abstract:Line 22 consider removing "model" or changing to "animal model" to distinguish from the computational model

Following the reviewer’s suggestion, we have modified the sentence to specify that we are using *C. elegans* is the animal model (line 22).

“To study how social information transmitted via pheromones can aid foraging decisions, we investigated the behavioral responses of the model animal *Caenorhabditis elegans* to food depletion and pheromone accumulation in food patches.”

Reviewer #4:The authors use the nematode *C. elegans* to reveal how animals associate social signals with specific contexts and modify their behaviors. Specifically, they show that *C. elegans* leaving a food patch are attracted to pheromonal cues, while those leaving later are repelled from pheromones. The authors using a behavioral model to suggest that the switch from attraction to repulsion is likely due to a change in learning. This study links learning with social signals providing a framework for further analysis into the underlying neuronal pathways.Line 36 + 46: The word decision is used, but lacks definition. Decision-making in *C. elegans* is controversial and should require precise language in using. If there is a decision to be made regarding exploitation of a food patch, what are the possible actions that the animal can choose to take?

We understand the reviewer’s concern about the controversies around the word “decision”. We decided to rephrase the paragraph and add a few sentences to explain the behavioral choices/actions available to an animal when feeding in a food patch that gives diminishing returns 32-48.

“Foraging for food is among the most critical activities for an animal's survival (Calhoun et al., 2014). It is also among the most challenging, because food is usually patchily distributed in space and time, and other individuals are attempting to find and consume the same resources (Abu Baker and Brown, 2014; Driessen and Bernstein, 1999; Stephens and Krebs, 1987).

An important factor, which has been the focus of considerable effort in models of foraging behavior, is for how long to exploit a food patch. At any given time, an individual feeding in a food patch has to choose between leaving to search for a better patch or staying. Leaving incurs the cost of exploring an unknown territory, while staying results in the cost of feeding in a depleting food patch. Most models addressing this “dilemma” involve patch assessment by individuals and postulate that the leaving time depends on local estimates of foraging success (Charnov, 1976; Oaten, 1977; Stephens and Krebs, 1987). As such, foragers are predicted to depart from a food patch when the instantaneous intake rate drops below the average intake rate expected from the environment, a phenomenon that has been observed in several animals, from insects (Wajnberg et al., 2008) to birds (Cowie, 1977; Krebs et al., 1974) and large mammals (Searle et al., 2005). The presence of other animals, however, affects individual foraging success so that different leaving times can be expected (Aubert-Kato et al., 2015; Couzin et al., 2005; Giraldeau and Caraco, 2000; Karpas et al., 2017).”

In addition, in the extended description of the model, we put these actions in the specific context of *C. elegans* feeding in a patchy environment (lines 134-145).

“Worms have three possible choices: (1) remain in their current food patch, (2) switch to another occupied food patch, and (3) disperse away from the occupied patches, in search for an unoccupied one. Switching means that a worm will leave its initial food patch and follow pheromone cues in order to find another occupied food patch. Dispersing means that the worm will leave its current food patch and avoid pheromones to maximize the probability of finding an unoccupied food patch. We assume that unoccupied food patches are hard to find, because they are not marked by pheromones and may be on average further away. Therefore, dispersal will not be advantageous until the occupied food patches are nearly depleted. A proof of this result, which closely resembles the marginal value theorem (Charnov, 1976), can be found in the supplement; here we will simply assume that worms will not disperse until the occupied food patches are depleted. Therefore, initially the individuals will choose between remaining or switching.”

Line 58: Similarly, the term associative learning requires definition.

We added the definition of associative learning. Now the sentence (lines 59-62) reads:

Moreover, it is not clear if the ability to use associative learning—the capacity to learn and remember the features of the environment that are associated with positive or aversive stimuli (Ardiel and Rankin, 2010)—to change the valence of pheromones could improve foraging success.

Line 99: As stated in the rebuttal, MY1 was used as (1) it was deemed to be more natural/ethological and (2) it is known to respond to synthetic pheromones. It is unclear whether or not the use of N2 in these experiments would change the results. If these results do not hold in N2, this is of note because it could lead to interesting follow-up experiments aiming to identify the biology underlying the MY1-specific behavior. The use of MY1 is cause for concern in truly placing these results within the full body of *C. elegans* literature. It seems prudent that the use of this strain be further addressed in the discussion.

We see the point raised by the reviewer. Based from our initial exploration, N2 could change its response to pheromones. However, since we did not perform the experiments at low density and with the repellent, we could not conclusively distinguish between feeding status and associative learning in driving the inversion of the pheromone preference. Although it is likely that the results of the above-mentioned experiments with N2 agree with MY1 observations, we agree with the reviewer that at the moment it is prudent to discuss the use of MY1 as a possible limitation of the generality of this study (see below). It is worth highlighting, however, that several natural strains show responses to pheromones very similar to MY1, mitigating the limitations deriving from the absence of experiments with N2.

The paragraph in the discussion (lines 429-437) reads:

“In conclusion, our study establishes a link between learning and social signals, providing a framework for further analysis unravelling the neuronal origin of the observed behaviors. However, the experiments presented here were performed with the natural isolate MY1. Thus, it remains to be tested if the same responses occur in the canonical lab strain N2, whose social behavior has changed due to laboratory domestication (Sterken et al., 2015), and in other natural strains with a social life more similar to MY1 (Greene et al., 2016). Nonetheless, by working with a natural isolate rather than N2, we could provide insights into the ecological significance of the inversion in the preference for pheromones and respond to the pressing need to further our knowledge of *C. elegans* ecology (Petersen et al., 2015).”

How far do pheromones diffuse to (within the detectable limit of a *C. elegans*)? Is it clear that the decision to leave a patch is made without knowledge of other food patches? Or is it possible that the animals are receiving pheromonal cues about other patches even while residing on a food patch bathed in pheromones. This distinction seems important to both the conclusions of the paper and the model and are not discussed in the body of the text. Can the diffusion of pheromones be experimentally defined or modeled to support the assumptions of the foraging model?

While these questions are very interesting, we believe that diffusion of the pheromone between patches is not a leading factor in our experiments or model: We focus on the behavior while worms are inside food patches, and in this case the concentration of pheromones coming from other worms in the patch is much higher than that coming from anywhere else. Given that it’s impossible to tell the origin of a detected molecule of pheromone, the higher concentration from the focal patch will mask emissions from any other food patches, even if diffusion is strong.

For example, consider a point source of pheromone with rate P. In three dimensions, the concentration profile at steady state is c(r) = P / (4*pi*D*r), where D is the diffusion constant of the pheromone and r is the distance to the source of the pheromone (e.g., the center of the occupied patch). We see that within a patch the concentration (and gradient) is always dominated by the local patch rather than a distant patch so long as the worm is closer to the patch that it is on rather than to any other patches. For example, the concentration of pheromone from the local patch is larger by a factor that is equal to the ratio of the distances, which will typically be larger than an order of magnitude. The ratio of gradients caused by the two patches is equal to the square of the ratio of distances, which is many orders of magnitude. Note that these conclusions are independent of the diffusion constant of the pheromone.

Is this foraging model ethological? Would worms not have a much lower payoff of leaving their patch in their natural boom-and-bust environment? Specifically, would patch density not be significantly lower in a natural environment such that switching patches is not beneficial until much later in food depletion (which could presumably be generations later)? Consider the spatial scale of a *C. elegans* to a rotting fruit.

From what we know about the ecology/ethology of *C. elegans*, it appears that adults do not travel from one apple to another because this type of dispersal depends on the dauer larva (see for example Frezal and Félix, *eLife* 2015; 4:e05849. DOI: 10.7554/*eLife*.05849). Therefore, in our model we are considering what happens, for example, inside one rotten apple, where food patches are bacterial colonies growing in different points of the apple. Patches on the same apple can have high density, get depleted over a timescale of hours or days, and travel between them occurs over a timescale of minutes or hours. This would justify the low cost associated with switching between food patches. We have clarified this in lines 139-141 of the revised manuscript.

“We assume that unoccupied food patches are hard to find, because they are not marked by pheromones and may be on average further away. Therefore, dispersal will not be advantageous until the occupied food patches are nearly depleted.”

Line 218: This brings up an interesting discussion point. The model and experiments assume stationarity of the pheromone blend over time with an inversion of valence occurring due to associative learning and satiety. However, would the pheromone blend not change throughout the course of the experiment? Could the specific combination of pheromones in the blend possibly be cause for the valence change?

In all the conditioning treatments in which we are adding the pheromone blend we can safely assume stationarity. The pheromone blend comes from a 9-day high-density liquid culture and thus is saturated with pheromones. Given that, the amount of pheromone leaked by the individual worms during the conditioning period (5 hours) is negligible and should not alter the pheromone blend. Accumulation of a pheromone cocktail different from the pheromone blend in the treatments without the addition of the pheromone blend was the reason why we performed the experiments at low worm density. However, temporal and spatial changes in the pheromone cocktail might be relevant in nature. Indeed, we cannot exclude that the composition of the pheromone cocktail plays a role in its valence as we state in the Discussion section at lines 413-420.

“The ability to assign a positive or negative preference to the pheromone blend through associative learning might depend also on other byproducts of worm metabolism or derive from the presence of multiple ascaroside molecules acting synergistically (Srinivasan et al., 2012). More studies are required to establish a link between associative learning and the composition of the pheromone blend, which is known to vary among developmental stages (Kaplan et al., 2011), sexes (Izrayelit et al., 2012) and strains (Diaz et al., 2014), ultimately allowing the discovery of novel roles for *C. elegans* pheromones (Viney and Harvey, 2017).”

Overall, I think this paper offers an interesting conclusion regarding optimal foraging and chemosensory valence. However, I do think it would benefit from precision of language and better discussion of the assumptions of the model and experimental conclusions.

We thank the reviewer for the valuable comments/ points of discussions and for appreciating our work.

[Editors' note: further revisions were suggested prior to acceptance, as described below.]

The paper offers an interesting conclusion regarding optimal foraging and chemosensory valence that is mostly supported by the data. However, the modelling part is comparatively weaker. Below is a list of the issues that still need to be addressed:– How reasonable is the assumption that all worms start in all patches at the same time? Naively, it would seem more realistic that a small founder population would arrive at a patch, and that these foraging dynamics would play out in a context that involves not only food depletion but also population growth. Would the main conclusions of the model (about optimal foraging and evolutionary stability) still hold in a more realistic model that considers such expected natural population dynamics?– How far do pheromones diffuse to (within the detectable limit of a *C. elegans*)? Is it clear that the decision to leave a patch is made without knowledge of other food patches? Or, is it possible that the animals are receiving pheromonal cues about other patches even while residing on a food patch bathed in pheromones. This distinction seems important to both the conclusions of the paper and the model, and is not discussed in the main text. Can the diffusion of pheromones be experimentally defined or modeled, to support the assumptions of the foraging model?– Is this foraging model ethological? Would worms not have a much lower payoff of leaving their patch in their natural boom-and-bust environment? Specifically, would patch density not be significantly lower in a natural environment such that switching patches is not beneficial until much later in food depletion (which could presumably be generations later)?– The model and experiments assume stationarity of the pheromone blend over time with an inversion of valence occurring due to associative learning and satiety. However, would the pheromone blend not change throughout the course of the experiment? Could the specific combination of pheromones in the blend possibly be cause for the valence change?– Finally, we wish to draw your attention to the STRANGE framework for animal behaviour research, which is currently being adopted by eLife and seems relevant to your study: https://doi.org/10.1038/d41586-020-01751-5

We thank the editor for highlighting the STRANGE framework to us.

Experience is listed among the STRANGE-related biases affecting the generality of behavioral studies. Indeed, a central finding in our work is that *C. elegans’* responses to pheromones depend on past experience and food and therefore should not be considered innate (as stated in the literature as a general finding). We have added a reference to STRANGE in the paragraph about this statement (lines 428-435 in the discussion).

“As a final remark, our results suggest that *C. elegans* preference for pheromones might not be innate, as it was previously stated (Greene et al., 2016; Macosko et al., 2009; Pungaliya et al., 2009; Simon and Sternberg, 2002; Srinivasan et al., 2012, 2008) and question what it means to be a naive worm (see also Webster and Rutz, 2020). Worms that we call “naive” are directly assayed for chemotaxis after being simultaneously exposed to both bacterial food and ascaroside pheromones, which are continuously excreted by the animals during their growth (Kaplan, 2011). Hence, it is possible that the attraction that “naive” worms exhibit is due to the positive association with food that they learn to make during growth on the plate.”

Reviewer #1:I am now satisfied with the improved presentation of model details.I do note a typo in the caption of Figure 2: "During the first phase, worms equalize occupancy the occupied patches."

Thanks. We have fixed it.

Reviewer #4:Gore et al., show that *C. elegans* leaving food patches at different times and different preferences to pheromones. Animals leaving early are attracted, while those leaving later are repelled from pheromones. I have no concerns with the data. Most of my concerns are with the model.It remains unclear whether the inversion is due the association of pheromones with food odorants or due to food-deprivation. I would recommend either testing these two possibilities or clarifying this in the manuscript with specific predictions for each outcome.

We apologize for not having stated this more clearly in the previous version of the manuscript. Our theoretical results show what features of *C. elegans’* environment might lead to the emergence of the observed behaviors, while remaining agnostic to the mechanism underpinning the behaviors. That is, the observed behaviors could have been triggered by associative learning or starvation, as hypothesized, as well as other factors. We used the experiments to establish which mechanism is the most plausible.

We added a paragraph at lines 196-200, which reads:

“Our theoretical results show what features of *C. elegans* environment may lead to the evolution of the observed behaviors, regardless of how the behaviors are implemented. In particular, the inversion in pheromone preference may be triggered by several different factors, and our model cannot distinguish between them. In the following, we will examine experimental evidence related to these mechanisms.”

Line 192 – Evolutionary Stable Strategy is introduced for the first time, but is neither defined nor cited. Further, small typo exists with the word "the" on this line.

We have added a definition of the term Evolutionary Stable Strategy following Maynard Smith at lines 130-133 when we introduce the model.

This model uses the tools of Game Theory to find the strategy that maximizes the food eaten by a worm, taking into account that other worms will also follow the same strategy. This strategy is called Evolutionary Stable Strategy, and should have been selected by evolution (Maynard Smith, 1982).

We have corrected the typo at line 195.